# Circadian oscillations in *Trichoderma atroviride* and the role of core clock components in secondary metabolism, development, and mycoparasitism against the phytopathogen *Botrytis cinerea*

**Marlene Henríquez-Urrutia**[1,2]**, Rebecca Spanner**[1,2]**, Consuelo Olivares-Yánez**[1,3]**, Aldo Seguel-Avello**[1,2]**, Rodrigo Pérez-Lara**[1,2]**, Hector Guillén-Alonso**[4]**, Robert Winkler**[4]**, Alfredo Herrera-Estrella**[5]**, Paulo Canessa**[1,3]*****, Luis F Larrondo**[1,2]*****

[1]ANID – Millennium Science Initiative Program - Millennium Institute for Integrative Biology (iBio), Santiago, Chile; [2]Pontificia Universidad Católica de Chile, Biological Sciences Faculty, Molecular Genetics and Microbiology Department, Santiago, Chile; [3]Centro de Biotecnología Vegetal, Facultad de Ciencias de la Vida, Universidad Andrés, Santiago, Chile; [4]Department of Biotechnology and Biochemistry, Cinvestav Unidad Irapuato, Irapuato, Mexico; [5]Laboratorio de expresión génica y desarrollo en hongos, Unidad de Genómica Avanzada-LANGEBIO, Irapuato, Mexico

***For correspondence:**
paulo.canessa@unab.cl (PC);
llarrondo@bio.puc.cl (LFL)

**Abstract** Circadian clocks are important for an individual's fitness, and recent studies have underlined their role in the outcome of biological interactions. However, the relevance of circadian clocks in fungal–fungal interactions remains largely unexplored. We sought to characterize a functional clock in the biocontrol agent *Trichoderma atroviride* to assess its importance in the mycoparasitic interaction against the phytopathogen *Botrytis cinerea*. Thus, we confirmed the existence of circadian rhythms in *T. atroviride*, which are temperature-compensated and modulated by environmental cues such as light and temperature. Nevertheless, the presence of such molecular rhythms appears to be highly dependent on the nutritional composition of the media. Complementation of a clock null (Δ*frq*) *Neurospora crassa* strain with the *T. atroviride*-negative clock component (*tafrq*) restored core clock function, with the same period observed in the latter fungus, confirming the role of *tafrq* as a *bona fide* core clock component. Confrontation assays between wild-type and clock mutant strains of *T. atroviride* and *B. cinerea*, in constant light or darkness, revealed an inhibitory effect of light on *T. atroviride's* mycoparasitic capabilities. Interestingly, when confrontation assays were performed under light/dark cycles, *T. atroviride's* overgrowth capacity was enhanced when inoculations were at dawn compared to dusk. Deleting the core clock-negative element FRQ in *B. cinerea*, but not in *T. atroviride*, was vital for the daily differential phenotype, suggesting that the *B. cinerea* clock has a more significant influence on the result of this interaction. Additionally, we observed that *T. atroviride* clock components largely modulate development and secondary metabolism in this fungus, including the rhythmic production of distinct volatile organic compounds (VOCs). Thus, this study provides evidence on how clock components impact diverse aspects of *T. atroviride* lifestyle and how daily changes modulate fungal interactions and dynamics.

## Editor's evaluation

The article by Henríquez-Urrutia and colleagues aims to establish that the fungus *Trichoderma atroviride*, a major biocontrol agent, has a circadian clock and test if this clock plays a role in development, secondary metabolite production, and mycoparasitism. The study is an important contribution toward the development of a new system for understanding how circadian rhythms influence fungal–fungal interactions that rests on solid experimental evidence.

## Introduction

*Trichoderma* species are cosmopolitan soil fungi found in several substrates, including decaying wood and fungal fruiting bodies, establishing beneficial interactions with plants (*Carreras-Villaseñor et al., 2012*; *Kubicek et al., 2019*; *Vos et al., 2015*), and negative ones with other microorganisms (*Druzhinina et al., 2006*; *Guzman-Guzman et al., 2019*; *Harman et al., 2004*; *Sood et al., 2020*). *Trichoderma* species are well-known mycoparasites as they can degrade, grow on other fungi, and even penetrate them, a distinctive feature that has been exploited in agriculture for their use as biocontrol agents of different phytopathogens, including *Botrytis cinerea* (*Naglot et al., 2015*; *Rahman et al., 2009*).

*Trichoderma*'s mycoparasitic behavior starts with the secretion of low levels of cell wall degrading enzymes (CWDE) (*Druzhinina et al., 2011*; *Seidl et al., 2009*; *Vos et al., 2015*) that dismantle the fungal prey's hyphae. Release of small molecules derived from the host can function as molecular signals in *Trichoderma*, allowing prey recognition (*Zeilinger et al., 2005*), redirecting hyphal growth, and initiating coiling around the host mycelium, followed by hyphal penetration. Indeed, the synergistic effect of enhanced CWDE and toxic secondary metabolite (SM) production allows *Trichoderma*'s hyphae to penetrate the prey's lumen, achieving its killing (*Belanger et al., 1995*; *Druzhinina et al., 2011*; *Sharma, 2011*; *Steyaert et al., 2003*).

*Trichoderma* species present developmental responses to different stresses, of which conidiation induced by light and mechanical damage has been extensively studied (*Casas-Flores and Herrera-Estrella, 2013*). Physical damage of mycelia in *Trichoderma atroviride* triggers a developmental injury response, leading to conidia production in the injured area through the generation of reactive oxygen species (ROS) and activation of MAPK signaling pathways (*Hernández-Oñate et al., 2012*; *Medina-Castellanos et al., 2014*). Additionally, light is also perceived as a stress signal, triggering conidiation: a light pulse (LP) given to a dark-grown *T. atroviride* generates several biochemical changes driving the formation of a green ring of conidia at the edge of the colony (*Casas-Flores et al., 2004*; *Casas-Flores et al., 2006*; *Galun and Gressel, 1966*; *Schmoll et al., 2010*). In *T. atroviride*, two genes called blue light regulators 1 and 2 (*blr1* and *blr2*) were identified as responsible for light-induced conidiation. The BLR proteins are homologs of the transcription factors WC-1 and WC-2, which in *Neurospora crassa* form the white collar complex (WCC), responsible for light and circadian regulation in this fungus (*Casas-Flores et al., 2004*; *Froehlich et al., 2002*; *Montenegro-Montero et al., 2015*). Importantly, light can be directly perceived thanks to the presence of a light-oxygen-voltage (LOV) domain in WC-1/BLR1 (*Corrochano, 2019*). Furthermore, BLRs have been described as critical for the effect of light on growth, oxidative stress responses, and light-controlled gene expression (*Cervantes-Badillo et al., 2013*; *Esquivel-Naranjo et al., 2016*; *Friedl et al., 2008a*; *Garcia-Esquivel et al., 2016*). Studies on BLR1 and BLR2 and the presence of a *frq* homolog that is induced by light, and whose expression depends on BLR1 (*Garcia-Esquivel et al., 2016*; *Steyaert et al., 2010a*), suggest the presence of a functional clock and underlying circadian regulation in this fungus. However, to date, neither circadian oscillations of *tafrq*/TaFRQ levels nor circadian phenotypes have been reported in *T. atroviride*; meanwhile, various studies have failed at detecting overt or molecular circadian rhythms in several fungi across fungal clades (*Cascant-Lopez et al., 2020*; *Hevia et al., 2016*; *Larrondo and Canessa, 2019*).

Circadian rhythms have emerged as an adaptation to Earth's rotation around its axis, allowing organisms to keep track of time, anticipating and adapting to periodic and predictable environmental changes through the temporal coordination of gene expression, metabolic pathways, physiological responses, and even behaviors (*Bell-Pedersen et al., 2005*; *Hurley et al., 2016*). Recent findings have shown that interactions between organisms may also be subjected to circadian regulation (reviewed in *Hevia et al., 2016*; *Larrondo and Canessa, 2019*). And while several reports have highlighted the

importance of the plant clock in plant–pathogen interactions, there has been scarce information from the pathogen perspective. Yet, it has been reported that the interaction between the phytopathogenic fungus *B. cinerea* and *Arabidopsis thaliana* largely depends on the fungal functional circadian clock, with virulence reaching higher levels at nighttime (*Hevia et al., 2015*; *Ingle et al., 2015*).

The ascomycete *N. crassa* is a salient model in chronobiology due to its robust circadian phenotype of daily spore production at dawn and a wealth of molecular resources (*Baker et al., 2012*). Its central oscillator is based on a negative transcriptional–translational feedback loop (TTFL), a shared core architecture across phyla (*Dunlap, 1999*; *Dunlap and Loros, 2017*; *Loros, 2020*). In *N. crassa*, the WCC activates the expression of the TTFL-negative element, *frequency* (*frq*), which leads to increasing levels of the FRQ protein, that starts recruiting kinases and negatively regulates its own expression by phosphorylating and inactivating the WCC. FRQ itself is progressively modified until reaching its maximum phosphorylation state at nighttime, losing its affinity for WCC and allowing a new transcription/translation cycle to start. Both *frq* mRNA and FRQ protein levels oscillate during this process with a period close to 22.5 hr (*Larrondo et al., 2015*; *Montenegro-Montero et al., 2015*). Albeit *wc-1*, *wc-2*, and *frq* homologs are present in several fungi, the description of overt clock phenotypes, as the molecular characterization of circadian rhythms in fungal species besides *N. crassa*, has been limited (*Montenegro-Montero et al., 2015*; *Salichos and Rokas, 2010*). Indeed, despite different studies addressing the latter, there are seldom examples of characterized circadian oscillators in other fungi and few studies exploring fungal circadian phenomena and clock components. Among these is *B. cinerea*, where rhythms in BcFRQ1 levels were confirmed, along with BcFRQ1's key role in conferring the daily difference on virulence potential. Remarkably, the evidence also suggests that in *B. cinerea* BcFRQ1 plays an extra-circadian role in developmental decisions as Δ*bcfrq1* displays altered developmental phenotypes observed even under conditions where the clock function is abolished, such as constant light, and in contrast to the phenotype observed in the absence of BcWCL1 (*Hevia et al., 2015*).

To deepen the role of circadian regulation in organismal interactions, and simultaneously evaluate the presence of a functional circadian clock in *T. atroviride*, we sought to assess the daily oscillations of TaFRQ expression under free-running conditions while assessing canonical clock properties and the clock's ability to respond to environmental stimuli. By generating clock mutant strains, we evaluated the impact of core clock components in *T. atroviride* secondary metabolism, developmental programs, and on its mycoparasitic capabilities against different *B. cinerea* strains, including those deficient in circadian regulation. Overall, our results reveal the presence of a functional circadian clock in *T. atroviride*, a role of core clock components in the modulation of mycoparasitic interactions, and extra-circadian functions for TaFRQ.

## Results

### *T. atroviride* bears a functional circadian clock

We generated TaFRQ^LUC, a translational fusion reporter between luciferase and TaFRQ (*Figure 1—figure supplement 1*) to assess oscillations of this putative *T. atroviride* core clock-negative element and quantify bioluminescence levels under free-running conditions (DD). Monitoring LUC activity after 3 days of constant light (LL) or light-dark (LD) 12:12 hr entrainment cycles, at 25°C, failed to reveal oscillations in multiple tested growth media (*Figure 1—figure supplement 2*, *Supplementary file 1*) with only one exception: GYEC, where weak but distinguishable oscillations were observed (*Figure 1A*, *Figure 1—figure supplement 3A*). Under the rationale that culture media appeared to be critical for *T. atroviride's* oscillations and considering that the fungus also interacts with plants, we decided to modify the GYEC media by adding plant-derived material, ground peas (GYEC+ peas), observing that the quality of the oscillations greatly improved (*Figure 1B*, *Figure 1—figure supplement 3B*). Notably, the results suggest that these rhythms are highly dependent on the nutritional composition of the media. The period estimation of those oscillations by fast Fourier transform nonlinear least squares (FFT-NLS) using BioDare (*Zielinski et al., 2014*) revealed an average period of 26.5 ± 0.36 hr. The phase of the first peak of TaFRQ expression occurs at DD 22, which corresponds to circadian time CT8, and is roughly similar (albeit slightly advanced) compared to the phase of FRQ^LUC expression (CT11) described in *N. crassa* (*Larrondo et al., 2012*). We could also observe oscillations (although weaker ones) by resetting the clock with 3 days in LL (*Figure 1C*) and also with a short 20 min LP

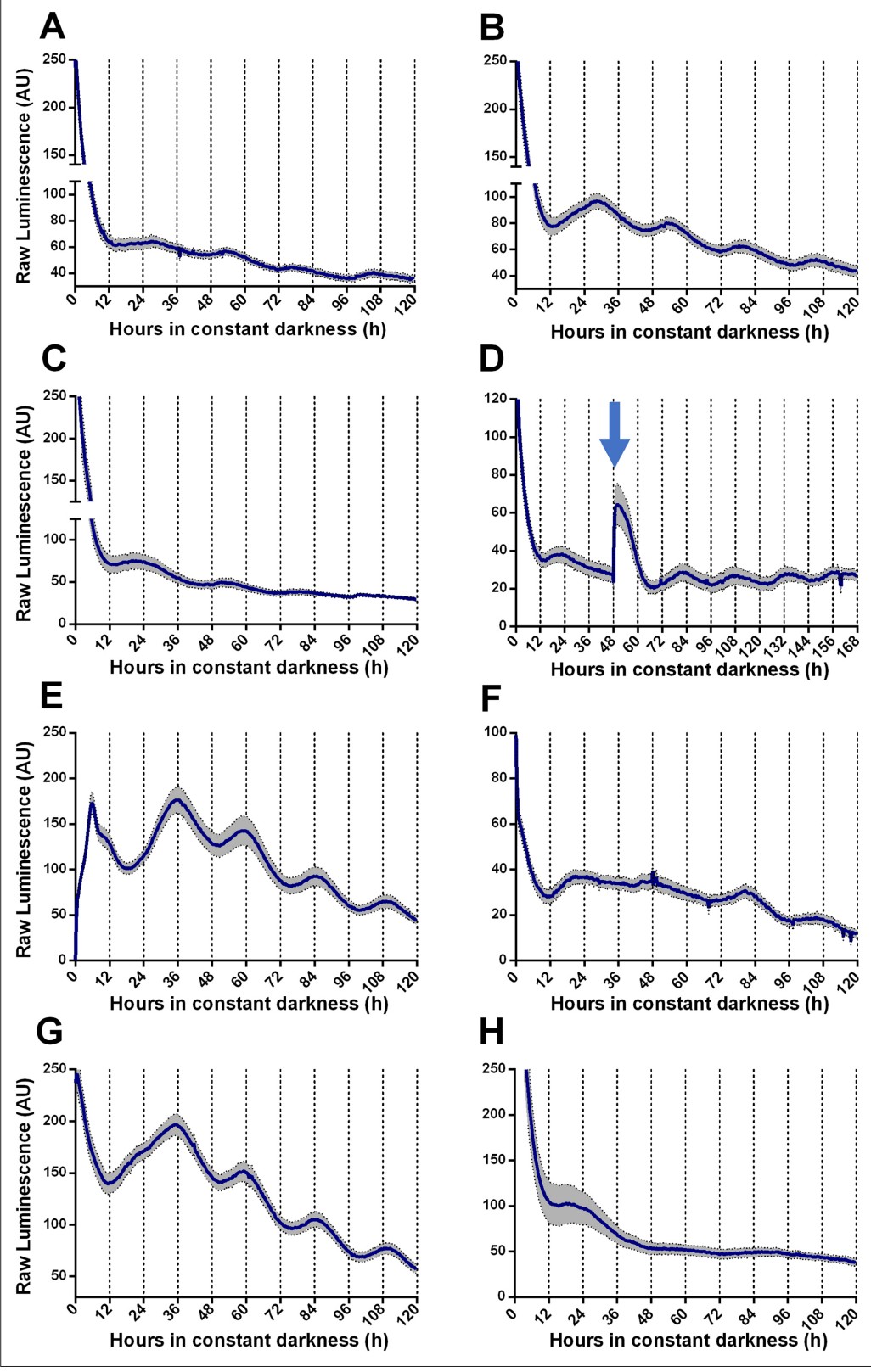

**Figure 1.** Circadian oscillations of TaFRQ$^{LUC}$ are reset by light and temperature and are temperature-compensated. (**A, B**) Raw TaFRQ$^{LUC}$ levels were quantified under DD in strains previously entrained for 3 days under LD (12:12 hr) cycles in GYEC (**A**) or GYEC+ peas (**B**) culture media. (**C**) Culture in LL (at 25°C) for 72 hr and then transferred to DD exhibit oscillations. (**D**) The effect of a 20 min. light pulse (blue arrow) was evaluated in DD on cultures that

*Figure 1 continued on next page*

*Figure 1 continued*

were also entrained in LD cycles as mentioned. (**E**) A 4°C temperature pulse efficiently resets rhythms, as shown in cultures grown for 48 hr in LL, cold pulsed in the dark for 6 hr, and then monitored in DD. (**F–H**) TaFRQ$^{LUC}$ expression was monitored at 20 (**F**), 22 (**G**), and 28°C (**H**) to assess rhythmicity at different temperatures. GYEC+ peas media was used in all cases except in (**A**), and bioluminescence was recorded at 25°C except when indicated otherwise. In all cases, raw data's mean values are presented from two biological and eight technical replicates each (SEM represented with the gray area). Source data can be found in *Figure 1—source data 1*.

The online version of this article includes the following source data and figure supplement(s) for figure 1:

**Source data 1.** For each graph, source data of TaFRQLUC bioluminescence is provided as an .xlsx file.

**Figure supplement 1.** Generation of TaFRQ$^{LUC}$ strains.

**Figure supplement 1—source data 1.** Related to *Figure 1—figure supplement 1B*.

**Figure supplement 1—source data 2.** Related to *Figure 1—figure supplement 1B*.

**Figure supplement 2.** TaFRQ$^{LUC}$ oscillations are conditioned by media composition.

**Figure supplement 3.** Normalized data of TaFRQ$^{LUC}$ oscillations in GYEC.

**Figure supplement 4.** A phase shift of TaFRQ$^{LUC}$ oscillations after different light entrainment and reset conditions.

**Figure supplement 5.** Oscillations of TaFRQ$^{LUC}$ protein levels in DD during a 52 hr time course.

**Figure supplement 5—source data 1.** Related to *Figure 1—figure supplement 5* (top).

**Figure supplement 5—source data 2.** Related to *Figure 1—figure supplement 5* (top).

**Figure supplement 5—source data 3.** Related to *Figure 1—figure supplement 5* (top).

**Figure supplement 5—source data 4.** Related to *Figure 1—figure supplement 5* (top).

**Figure supplement 6.** TaFRQ$^{LUC}$ oscillations observed at different temperatures demonstrate temperature compensation in *T. atroviride*.

**Figure supplement 7.** Temperature compensation in TaFRQ$^{LUC}$.

**Figure supplement 8.** Evaluation of transcriptional luciferase reporters in *T. atroviride*.

---

after 48 hr in DD, which elevates TaFRQ$^{LUC}$ levels, boosting sustained oscillations for at least 4 days (*Figure 1D*). Likewise, a low-temperature pulse of 4°C for 6 hr also served as a cue to reset the clock, shifting its phase almost 12 hr, similar to what can be seen in *N. crassa* when monitoring FRQ$^{LUC}$ expression (*Larrondo et al., 2012*; *Figure 1E*). After both pulse protocols, the period also remained in the 26 hr range (26.30 hr and 26.49 hr, respectively), exhibiting even better oscillations after the perturbation, as previously observed for clock reporters under some conditions (*Shi et al., 2007*). These findings indicate that circadian oscillations of TaFRQ can be entrained by LD cycles or shifted by light and temperature pulses. Indeed, the analyses of these perturbations (*Figure 1—figure supplement 4*) revealed a mean phase delay of 4.29 hr between LD and LP entrainment and 10.85 hr between LL reset and 4°C pulses, further showing that these oscillations behave as one would expect from a circadian oscillator (*Heintzen and Liu, 2007*). To complement these results, we used a Western blot with anti-LUC antibodies to obtain evidence of TaFRQ$^{LUC}$ oscillations in constant darkness over a 52 hr time course, further supporting the circadian variation of TaFRQ levels (*Figure 1—figure supplement 5*). Importantly, we also evaluated if the observed TaFRQ$^{LUC}$ oscillations were temperature-compensated, meaning that they should maintain about the same period in a physiological range of temperatures. Thus, TaFRQ$^{LUC}$ rhythms were assessed after 3 days in 12:12 LD entrainment regimes at 20, 22, 25, and 28°C and monitored in DD at the corresponding temperatures (*Figure 1B and F–H*, *Figure 1—figure supplement 6*). While low-amplitude oscillations were observed at 20°C, changes in TaFRQ$^{LUC}$ levels exhibited higher amplitude at 22°C than at 25°C, whereas oscillations were lost at 28°C, indicating that of the tested temperatures the *Trichoderma* clock runs better at 22°C. To quantify the degree of temperature compensation, we calculated the $Q_{10}$ between 20 and 25°C (*Figure 1—figure supplement 7*), obtaining a $Q_{10}$ of 1.05 ± 0.07, which is in the range for temperature compensation described in other circadian systems (*Anderson et al., 1985*; *Kusakina et al., 2014*; *Mattern et al., 1982*). Thus, the results show that the *T. atroviride's* TaFRQ$^{LUC}$ circadian oscillations can compensate in a temperature range between 20 and 25°C, further supporting that a circadian clock is responsible for the observed rhythms.

To test rhythmic regulation of putative clock-controlled genes (*ccgs*), we subjected *luc* expression under the control of four distinct *T. atroviride* promoters of interest. These were selected from homologs to *N. crassa* genes encoding trehalose synthase, glyceraldehyde 3-phosphate dehydrogenase, superoxide dismutase, and CON-10, all of which have been reported as rhythmic in *Neurospora* and some even in other organisms (*Lakin-Thomas et al., 2011*; *Lillo, 1993*; *Shinohara et al., 2002*; *Yoshida et al., 2008*). Although in all cases we detected bioluminescence, no oscillations were identified in DD, at least under the tested culture conditions (*Figure 1—figure supplement 8*, *Supplementary file 2*). Despite the disappointing lack of oscillation of these putative ccgs reporters, this result reinforces the idea that the rhythms yielded by the TaFRQ^LUC reporter strains are of a circadian nature rather than spurious oscillations driven by external cues or rhythms in ATP levels.

## Functional conservation of core clock elements between *T. atroviride* and *N. crassa*

In *N. crassa,* transcriptional activation of *frq* under constant darkness requires recognition, by the WCC, of the clock box (*c-box*) in its promoter. This *cis*-element is necessary and sufficient to generate and maintain *frq* daily rhythms (*Froehlich et al., 2003*). To evaluate the conservation of the circadian system of *T. atroviride,* we transformed this fungus with *luc* under the control of the *N. crassa c-box* (*Nc*c-box-luc). Notably, this reporter revealed clear rhythms of *luc* levels under free-running conditions (DD), which responded to light and temperature cues, and whose presence was also conditioned by culture media composition (*Figure 2*, *Figure 2—figure supplement 1*). The bioluminescent traces were analyzed using BioDare (*Zielinski et al., 2014*), revealing an average of 25.67 ± 1.01 hr period for two independent *Nc*c-box-luc strains, in close agreement with the TaFRQ^LUC data. This provides additional proof of a functional circadian clock in *T. atroviride* and demonstrates that a pivotal *Neurospora cis*-element (*Nc*c-box) can be recognized in *T. atroviride*. Albeit the measured period with this reporter is slighter shorter compared to the one obtained with TaFRQ^LUC, this also recapitulates what has been seen when comparing both types of reporters (translational and transcriptional) in *N. crassa*, which could occur due to discrete changes in FRQ structure/dynamics when analyzing this protein as a translational fusion (*Larrondo et al., 2012*). Notably, in response to an LP, a clear and acute augment in *luc* expression was seen, as observed for light-responsive elements in *Neurospora*, arguing that this *Nc*c-box could integrate light and clock information in *T. atroviride*. This hypothesis will be challenged in future experiments.

In order to provide additional and compelling evidence that the *T. atroviride* FRQ is a bona fide core clock component, we adopted a classic genetic approach, based on its ability to complement the lack of the endogenous FRQ protein in *N. crassa*. Thus, we complemented a *Neurospora Δfrq* strain, containing a circadian luciferase reporter (*his-3::frq_c-box-luc* reporter), with the *tafrq* gene in which its expression was under the control of the native promoter and terminator of *Neurospora* (*Figure 2—figure supplement 2A and B*). The presence of TaFRQ in this *Δfrq::tafrq* strain yielded clear rhythmic levels of bioluminescence, in contrast to full arrhythmic and high levels of LUC in the recipient *Δfrq* strain (*Figure 2D*). Most notably, the analyses of the rhythms revealed a period of ~26 hr after 3 days of 12:12 hr LD entrainment (*Figure 2E*), further confirming that the long free-running period described in *Trichoderma* is a property encoded in the FRQ sequence, providing clear evidence of interspecies conservation of a phospho-code commanding period (*Figure 2—figure supplement 3*). However, no overt rhythmic conidiation was observed in these strains (*Figure 2—figure supplement 4*), with or without LD entrainment, suggesting that whilst the core clock may function, the normal outputs of the clock in *N. crassa* may no longer be responding. Finally, to get a first glimpse of TaFRQ expression in *Neurospora*, we conducted a short 12 hr time course, confirming that after an LL to DD transfer this heterologous clock component is degraded as expected (*Figure 2—figure supplement 2C*).

## TaFRQ is required for proper conidiation in *T. atroviride* after light exposure and mechanical injury

While in *Neurospora* FRQ appears to solely act as a clock component, in *B. cinerea* BcFRQ1 exhibits extra-circadian roles, impacting developmental phenotypes. Indeed, enhanced microconidiation and sclerotia formation are observed in *Δbcfrq1* (but not in *Δbcwcl1*) even in conditions where a fungal circadian clock does not run such as constant light (*Hevia et al., 2015*). This phenotype, which contrasts WT *B. cinerea* and *Δbcwcl1* copious production of macroconidia in LL and LD, prompted

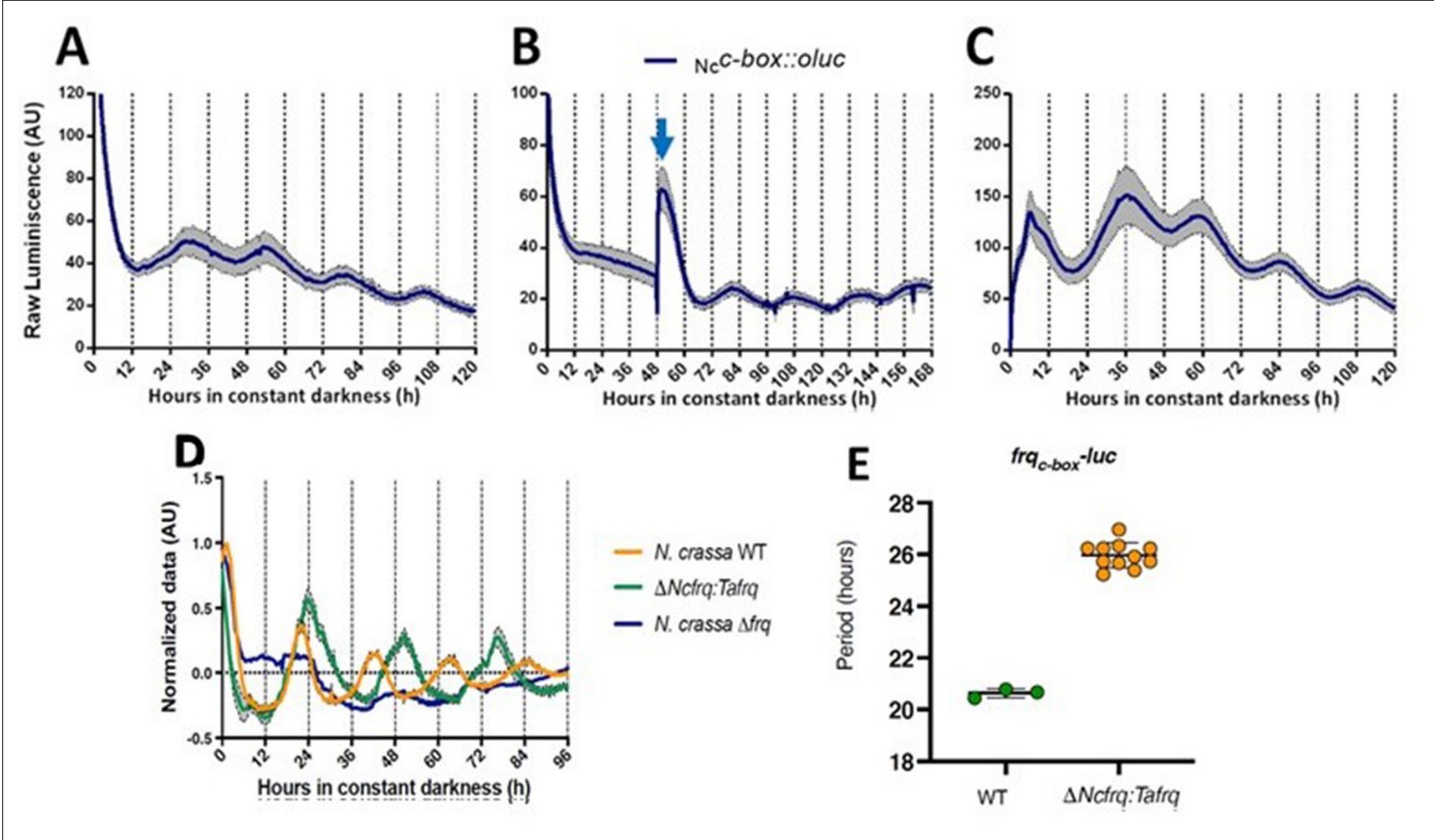

**Figure 2.** Functional conservation of circadian clock components between *T. atroviride* and *N. crassa*.
 (**A, B**) $_{Nc}$c-box-luc reporter strains were entrained for 3 days under LD (12:12 hr) cycles at 25°C and then transferred to DD where they were monitored unperturbed (**A**), or examined after a discrete light pulse (blue arrow), which produces a perceivable phase shift (**B**). (**C**) Strains were grown for 48 hr in LL and then transferred to DD, where a 4°C temperature treatment for 6 hr was applied. Then bioluminescence was monitored under constant darkness and 25°C. GYEC+ peas culture media was used in all cases. Raw data is presented with mean values of two biological replicates with three technical replicates each (SEM is represented with the gray area). (**D, E**) The expression of *tafrq* in *N. crassa* is able to restore circadian oscillation of a strain lacking its endogenous *frq*. The use of the transcriptional reporter *frq$_{c-box}$-luc* reveals the presence of circadian rhythms in luciferase in the *Δfrq::tafrq* strain where *Trichoderma* FRQ was expressed. These oscillations have a longer period than WT *N. crassa*. Period determination was performed using BioDare 2.0, Linear Det, FFT-NLLS algorithm. A significant difference in period was found between WT *N. crassa* strain and the strain carrying *T. atroviride* FRQ (Mann–Whitney test, p-value=0.0055). Period ± SD. *frq$_{c-box}$-luc*: WT: 20.6 ± 0,2; *ΔNcfrq::tafrq*: 25.9 ± 0.5. Source data can be found in *Figure 2—source data 1*.

The online version of this article includes the following source data and figure supplement(s) for figure 2:

**Source data 1.** Source data of bioluminescence for each graph is provided as an .xlsx file.

**Figure supplement 1.** Evaluation of $_{Nc}$c-box-luc luciferase reporters in *T. atroviride*.

**Figure supplement 2.** Generation of *Δncfrq::tafrq* strains in *N. crassa*.

**Figure supplement 2—source data 1.** Related to *Figure 2—figure supplement 2B*.

**Figure supplement 2—source data 2.** Related to *Figure 2—figure supplement 2B*.

**Figure supplement 2—source data 3.** Related to *Figure 2—figure supplement 2C* (top).

**Figure supplement 2—source data 4.** Related to *Figure 2—figure supplement 2C* (top).

**Figure supplement 2—source data 5.** Related to *Figure 2—figure supplement 2C* (bottom).

**Figure supplement 2—source data 6.** Related to *Figure 2—figure supplement 2C* (bottom).

**Figure supplement 3.** FRQ protein alignment between *N. crassa* and *T. atroviride*.

**Figure supplement 4.** Race tube analysis of four *Δncfrq::tafrq* strains to evaluate clock-controlled rhythmic conidiation.

us to ask whether *tafrq* might also serve a role in fungal development in *T. atroviride*. Therefore, we constructed *tafrq* deletions (*Δtafrq*) (*Figure 3—figure supplement 1*) and tafrq-overexpressing strains (OE::*tafrq*) (*Figure 3—figure supplement 2*) and evaluated light-induced conidiation for 7 days in LL. After 72 hr of cultivation in LL, *T. atroviride* displayed the characteristic green coloration as a result of regular conidia formation. In contrast, the *Δtafrq* strain exhibited a dark-green color not observed also in the OE::*tafrq* strains (*Figure 3A*). Notably, at 3 days post-inoculation (dpi), *Δtafrq* had produced fewer conidia, albeit this difference was no longer significant by day 7, indicating that conidiation in *Δtafrq* is delayed but not fully compromised. Furthermore, conidia formation in response to a discrete LP was significantly decreased in *Δtafrq,* regardless of the pulse duration (*Figure 3C*). In contrast, no significant differences were observed between TaWT and OE::*tafrq* in both types of analyses (*Figure 3B and C*). Importantly, although in the latter strain *tafrq* mRNA levels were not boosted dramatically, they were statistically higher compared to WT (*Figure 3—figure supplement 2*).

We also evaluated whether the injury-response mechanism that drives conidia formation in *T. atroviride* upon physical damage was affected in the mutant strains (*Hernández-Oñate et al., 2012*). We injured the WT and the aforementioned clock mutant strains in DD, including *Δblr1,* employing a sterile scalpel and quantified conidia formation 48 hr afterward. Interestingly, *Δblr1* showed a 51% average reduction in conidia production compared to the TaWT strain, which had not been previously reported. In contrast, while the OE::*tafrq* showed no significant differences compared to TaWT, *Δtafrq* exhibited almost a 100-fold reduction in injure-triggered conidiation (*Figure 3A and D*). As expected, we were able to restore the injury-response conidiation when complementing *Δtafrq* with *tafrq* back at its endogenous locus (*ΔtafrqC*) (*Figure 3—figure supplements 3 and 4*). Importantly, the results obtained in LL suggest that TaFRQ has an underlying role in promoting conidiation, providing evidence of an additional role for TaFRQ in *T. atroviride*, besides its expected main function as a negative clock element. A similar conclusion can be drawn about response to an acute stress stimulus, such as mechanical damage, since even though this was done in DD conditions, sporulation per se does not appear to be under clock control in *Trichoderma*.

## Light and core clock components regulate the interaction between *T. atroviride* and *B. cinerea*

To evaluate the role of *T. atroviride* core clock components in its mycoparasitic capabilities, we performed confrontations with *B. cinerea*, utilizing the corresponding clock mutant strains lacking *wc-1* or *frq* homologs (*Canessa et al., 2013*; *Hevia et al., 2015*). Experiments were first conducted under LL and DD conditions, evaluating the mycoparasitic behavior of *T. atroviride* as its ability to overgrow *B. cinerea*, as shown in *Figure 4A*. Initially, we performed confrontations on GYEC and GYEC+ peas medium, but as both fungi grew as white mycelium we could not differentiate their respective growth front, making the observation of the overgrowth phenotype difficult. Therefore, we decided to perform the confrontations on PDA. Importantly, before performing the confrontation experiments, we first analyzed whether the growth of both fungi was altered under the tested culture conditions or whether, when growing on the same Petri dish before the physical encounter, there were growth delays that could affect data interpretation. Results showed that growth rates of each strain (alone or interacting) were not affected in any of these scenarios (*Figure 4—figure supplement 1*).

In the TaWT–B05.10 *interaction, T. atroviride* displayed higher overgrowth over *B. cinerea* in DD than in LL (50% vs. 20%, respectively). Increased overgrowth in DD (90%) versus LL was also seen for *Δblr1* against B05.10, being the most aggressive of the tested *T. atroviride* strains. At the same time, *Δtafrq* displayed similar behavior to TaWT in LL and DD against B05.10 (*Figure 4B and C*, *Supplementary file 3*). The confrontation *of T. atroviride* strains with B. cinerea *Δbcfrq1* was similar to what was observed for B05.10 in LL and DD, except for the *Δtafrq–Δbcfrq1* interaction in LL, where no overgrowth was observed. Finally, under LL, *Botrytis Δbcwcl1* was easily overgrown by TaWT but not by the *Δblr1* or *Δtafrq Trichoderma* mutants*. However, in the absence of light, *Δbcwcl1* was easily overgrown by ~80% by all *Trichoderma* strains (*Figure 4B and C*, *Supplementary file 3*). Overall, these results suggest that constant light inhibits the overgrowth capacity of *T. atroviride* irrespective of the presence of *blr1*, or that mycoparasitism is enhanced in darkness. Nevertheless, whether other photoreceptors are involved in this response remains to be investigated. Core clock components also modulate the outcome of these mycoparasitic interactions, suggesting an inhibitory role of BLR1, as its absence enhances overgrowth in DD against all *B. cinerea* strains.

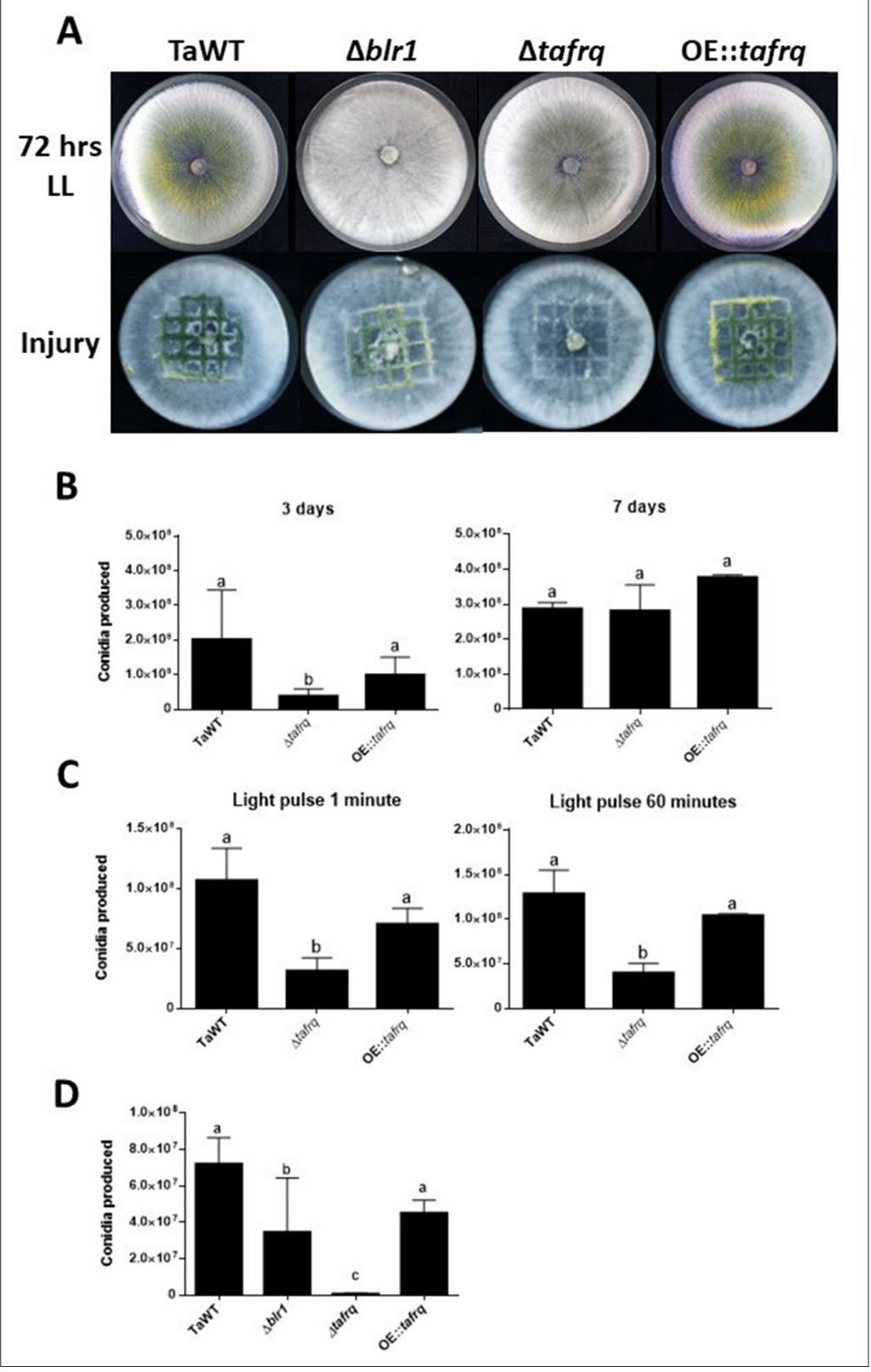

**Figure 3.** Effect of light and injury-induced in conidiation in clock component mutant strains of *T. atroviride*. TaWT, Δ*blr1*, Δ*tafrq*, and OE::*tafrq* strains were grown in potato dextrose agar (PDA) at 25°C. (**A**) Sporulation was observed at 72 hr in LL, or after injury (mechanical damage) in DD conditions, as specified in 'Materials and methods.' (**B**) Conidia produced after 3 and 7 days of growth in LL. (**C**) Conidia production of *Trichoderma*'s

*Figure 3 continued on next page*

*Figure 3 continued*

strains after receiving a 1 and 60 min light pulse (LP). LP was applied to cultures previously grown for 36 hr in DD. (**D**) Conidia produced after mechanical damage. Strains were grown for 36 hr in DD, and injury was performed with a scalpel using a grid as reference. Later, damaged strains were kept in DD for 48 hr to allow conidia collection. Errors bars represent the mean ± SD of three biologicals for (**B**) and four for (**C**) replicates for each experiment. Different letters indicate significant differences based on Tukey's test (p<0.05).

The online version of this article includes the following source data and figure supplement(s) for figure 3:

**Figure supplement 1.** Generation of *Δtafrq* strain in *T. atroviride*.

**Figure supplement 1—source data 1.** Related to *Figure 3—figure supplement 1B*.

**Figure supplement 1—source data 2.** Related to *Figure 3—figure supplement 1B*.

**Figure supplement 1—source data 3.** Related to *Figure 3—figure supplement 1C*.

**Figure supplement 1—source data 4.** Related to *Figure 3—figure supplement 1C*.

**Figure supplement 2.** Expression profile of *tafrq* in TaWT and OE::*tafrq* strains.

**Figure supplement 3.** *Δtafrq* complementation.

**Figure supplement 3—source data 1.** Related to *Figure 3—figure supplement 3B–D*.

**Figure supplement 3—source data 2.** Related to *Figure 3—figure supplement 3B–D*.

**Figure supplement 3—source data 3.** Related to *Figure 3—figure supplement 3E*.

**Figure supplement 3—source data 4.** Related to *Figure 3—figure supplement 3E*.

**Figure supplement 4.** Injury-induced conidiation is recovered in *Δtafrq*-complemented strains.

In contrast, *tafrq* does not seem to have a relevant role in the outcome of the *T. atroviride* interaction with B05.10, albeit the mutant has a weaker mycoparasitic capacity when interacting in LL with *Δbcwcl1 or Δbcfrq1*. Similarly, the absence of *B. cinerea bcfrq1* does not impact the interaction with TaWT. Still, the lack of *bcwcl1* increases the susceptibility of *B. cinerea* to *T. atroviride,* suggesting a diminished capacity to respond to this biotic stress in the mutant.

## Differential overgrowth capacity of *T. atroviride* against *B. cinerea* in light/dark cycles

To fully understand the role of core clock components in the interaction between *T. atroviride* against *B. cinerea*, we performed confrontations under circadian paradigm assays in LD 12:12 hr cycles. For this, strains were inoculated at 'dawn' when lights are turned on in the LD cycle (AM) or inoculated at 'dusk' before lights are turned off in the DL (PM) regime (*Figure 5A*). Confrontations were kept in their respective photoperiod cycles for 7 days, after which the degree of *Trichoderma* overgrowth over *B. cinerea* was evaluated for different strain combinations.

In TaWT–B05.10 interactions under LD cycles, *Trichoderma*'s overgrowth was enhanced in plates inoculated at dawn (AM) rather than at dusk (PM) (*Figure 5B and C*, *Supplementary file 3*). The *Δtafrq*–B05.10 interactions behaved similarly to TaWT-B05.10, suggesting that TaFRQ does not cause the differential mycoparasitic behavior seen in LD (enhanced capacity under the AM inoculation protocol). In contrast, *Δblr1* was capable of overgrowing WT *Botrytis* to the same extent in both AM and PM inoculation protocols, suggesting that indeed, BRL1 has a role in the differential outcome of interactions performed in LD cycles (*Figure 5B and C*, *Supplementary file 3*).

Centering the analysis on *B. cinerea*, *Δbcfrq1* was overgrown by all *Trichoderma* strains to the same extent in both cycles (AM or PM), revealing the disappearance of the clock-related phenotypes observed for B05.10. Thus, *frq* mutants in *Trichoderma* and *B. cinerea* display major differences regarding their behavior. While the absence of TaFRQ does not alter the time-of-the-day difference in mycoparasitic interactions (against B05.10 and *Δbcwcl1* strains), the lack of *BcFRQ1* does, suggesting a major role for the latter in the outcome of the daily interactions. Regarding *Δbcwcl1*, it mimics the differences seen for B05.10 but is more aggressively overgrown by *Trichoderma* in the AM protocol. In total, these results show that both core clock components (WC-1 and FRQ) and the time of the day in which the interaction is staged affects the outcome of the fungal-fungal interactions, where the *B. cinerea* element (BcFRQ1), but not the *T. atroviride* TaFRQ element, appears as critical.

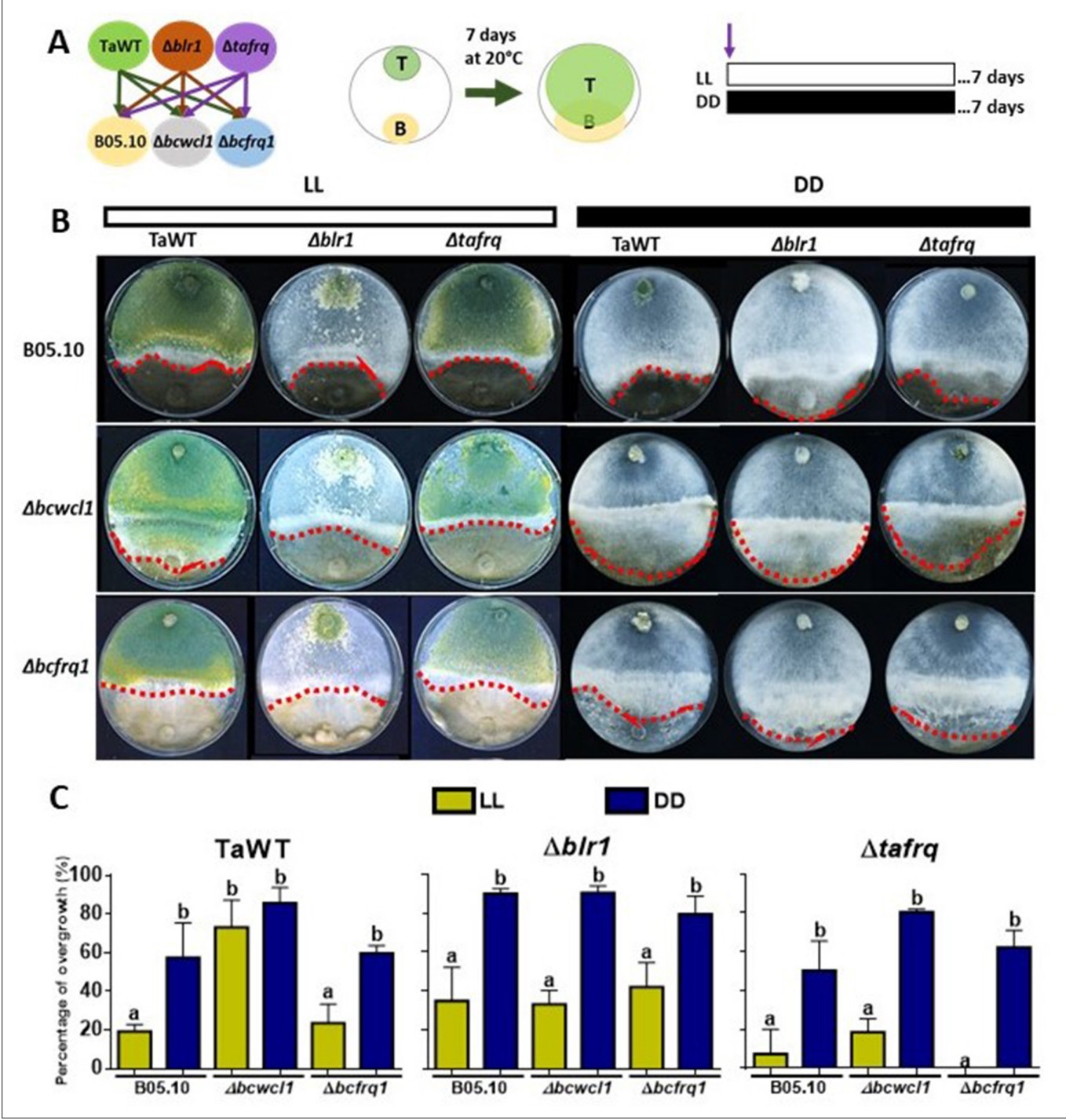

**Figure 4.** Core clock components, light, and darkness modulate the *T. atroviride* and *B. cinerea* interaction outcomes.
 (**A**) Schematic representation of the experimental design (left) and confrontation assay (right). (**B**) After 7 days of cultivation in LL and DD conditions in potato dextrose agar (PDA) culture media, *T. atroviride* overgrowth was evaluated. Cultures were kept at 20°C during 7 days under the corresponding light regimes. Red dotted lines indicate the edge of *Trichoderma* over *B. cinerea*. A representative image of three biological replicates is shown. (**C**) Percentage of *B. cinerea* colony area covered by *Trichoderma* (percentage of overgrowth). Error bars represent the means ± SD of three biological replicates. Different letters indicate significant differences based on Tukey's test (p<0.05).

The online version of this article includes the following figure supplement(s) for figure 4:

**Figure supplement 1.** Control of growth rates of *T. atroviride* and *B. cinerea* in confrontation assays conducted in potato dextrose agar (PDA) conditions.

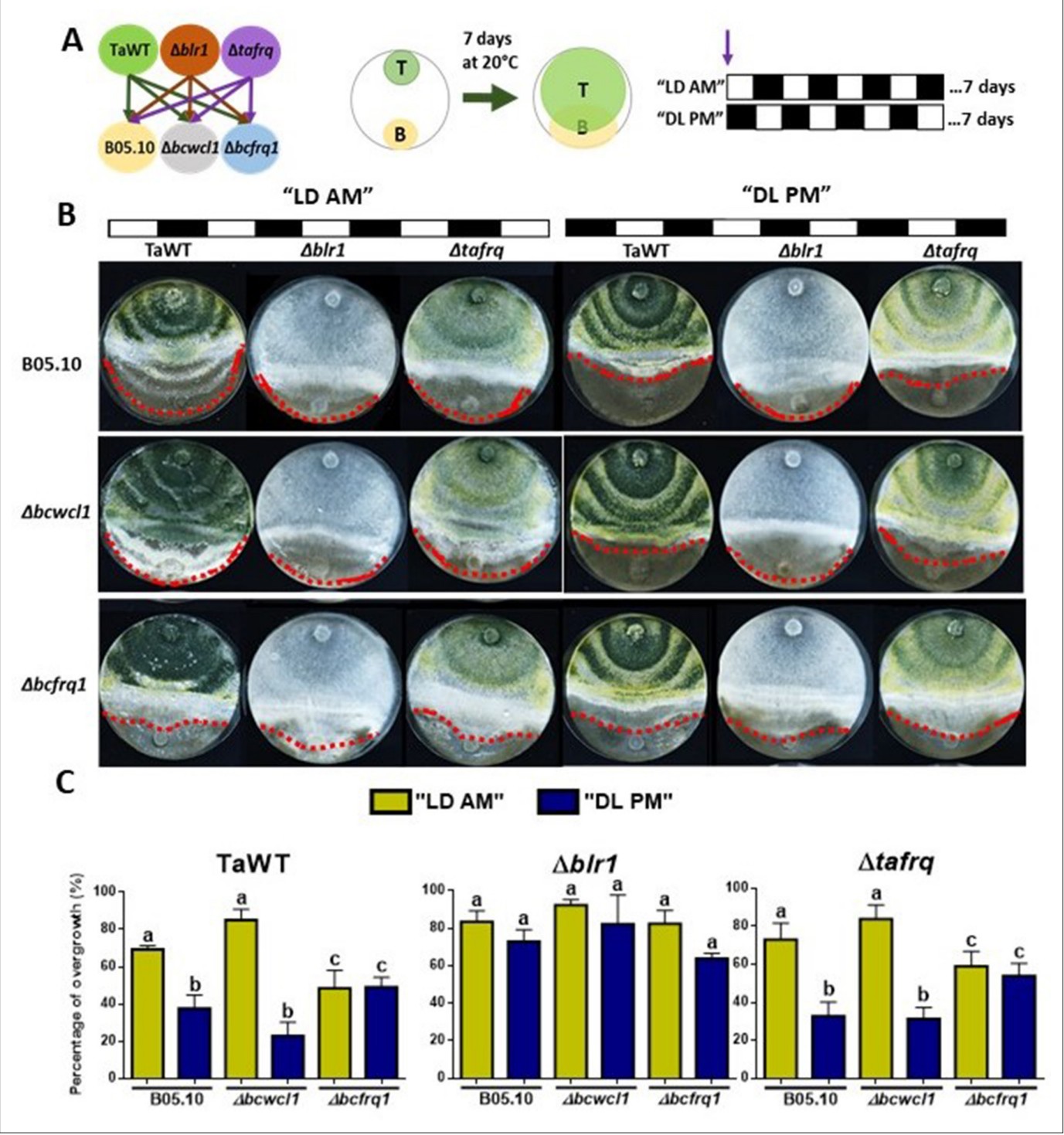

**Figure 5.** Core clock components of *B. cinerea* and *T. atroviride* have a role in the differential mycoparasitic interaction observed under LD cycles. (**A**) Schematic representation of the experimental design (left) and confrontation assays (right) in which fungal strains were inoculated in potato dextrose agar (PDA) at dawn, that is, at lights on in the light:dark (LD or AM) cycle or at lights off in the dark:light cycle (DL or PM regime). In both cases, cultures were kept at 20°C during 7 days under the corresponding light regimes. (**B**) After 7 days of cultivation, overgrowth by *T. atroviride* was evaluated: red dotted lines indicate the edge of the latter over *B. cinerea*. A representative image of three biological replicates is shown. (**C**) Percentage of *B. cinerea*'s colony area covered by *Trichoderma* (percentage of overgrowth). Error bars represent the means ± SD of three biological replicates. Different letters indicate significant differences based on Tukey's test (p<0.05).

## Secondary metabolism is affected in clock mutant strains of *T. atroviride*

*Trichoderma* species are prolific producers of SM that have been implicated in the effectiveness of mycoparasitism against pathogens (*Reino et al., 2008*). To identify whether the putative core clock components could affect SM production, we carried out a metabolic fingerprinting to obtain an overview of global SM produced by TaWT and core clock component mutant strains grown in PDA under LL, DD, and after a 4 hr LP.

Extracts of diffusible compounds were analyzed by direct injection to mass spectrometry. The production of 230 compounds was statistically different among strains and conditions (ANOVA p<0.05). The overview of diffusible compounds produced by TaWT, Δ*blr1*, Δ*tafrq,* and OE::*tafrq* in DD, LL, and LP (*Figure 6A*) indicates that production of diffusible compounds is particularly enhanced in Δ*tafrq* in all conditions compared to TaWT, Δ*blr1*, and OE::*tafrq*. In TaWT, 103 and 15 molecules showed a twofold induction in LL and LP compared to DD, respectively, which suggests that light induces the production of diffusible compounds in *T. atroviride*. A comparison of diffusible compound levels between TaWT and Δ*blr1* showed that 129 molecules were at least twofold reduced in LL in the mutant strain. Many compounds were induced after an LP and a few others in DD (*Table 1*).

Among the diffusible compounds, the molecule 1679.08 m/z had an interesting behavior between strains and light conditions. In TaWT, this molecule showed the greatest fold induction in LL and LP compared to DD. On the contrary, in Δ*blr1,* its production was reduced in LL, but it was high in DD compared to TaWT (*Figure 6B*). This suggests that BLR1 is necessary for light induction of several diffusible compounds in TaWT, including 1679.08 m/z, as their production is severely dampened in the Δ*blr1* mutant in LL. However, it seems that the production of a subset of diffusible molecules is either repressed in DD by BLR1 and/or additional photoreceptors are involved in light induction of diffusible molecules after an LP in Δ*blr1*.

As mentioned, diffusible compounds are enhanced in Δ*tafrq,* with several molecules showing at last a twofold induction compared to TaWT. The levels of some, like 381.77 m/z, appear downregulated in TaWT and Δ*blr1* but enhanced in Δ*tafrq* (*Figure 6A and B*, *Table 1*). This suggests that TaFRQ represses the production of diffusible compounds under these conditions. On the contrary, TaFRQ could also have a positive role in the light-mediated biosynthesis of molecules such as 1679.08 m/z, which is one of the few downregulated compounds in Δ*tafrq* in LL. Despite our findings, we did not observe a correlation between enhanced production of diffusible compounds with enhanced overgrowth of *Trichoderma* strains as Δ*tafrq* has a weak overgrowth behavior (*Figure 4*). Finally, levels of the detected compounds in OE::*tafrq* are also high in the three experimental conditions compared to TaWT but less than in Δ*tafrq* (*Table 1*). This suggests that the absence or misregulation of clock components causes an upregulation of SM, reinforcing the idea of extra-circadian roles for TaFRQ in secondary metabolism.

These analyses show that light induces the production of a group of diffusible compounds and that BLR1 and TaFRQ have distinct (opposite) roles in regulating such SMs.

As diffusible molecules represent only part of the metabolic complexity of *Trichoderma*, we also analyzed the profile of volatile organic compounds (VOCs) by measuring their production in vivo with a low-temperature plasma ionization MS (LTP-MS) under the same previously described conditions. The analyses revealed that the production of 131 compounds (in a 50–500 m/z range) was statistically different across all strains and growth conditions (ANOVA p<0.05). A global view of VOCs reveals an enhanced production in LL in Δ*blr1* and Δ*tafrq,* with a group of VOCs mainly induced in Δ*blr1* and reduced in Δ*tafrq*. On the contrary, in DD, many volatile compounds were upregulated in Δ*tafrq* and downregulated in Δ*blr1* (*Figure 6C*).

As volatile compounds showed higher dispersion and a narrow range of variation, we analyzed molecules that were statistically different across strains and culture conditions, leading to over 1.5-fold change between TaWT and mutant strains. Comparison between Δ*blr1* and TaWT detected light-induced as well as dark-repressed compounds (*Table 1*) in Δ*blr1*. In Δ*tafrq*, several compounds were induced in DD and LL compared to TaWT, but also some compounds appeared as repressed in all three conditions for Δ*tafrq* (*Table 1*). In Δ*tafrq*, the compound 371.03 m/z had the highest production in DD compared to TaWT (*Figure 6D*). No significant differences were found between OE::*tafrq* and TaWT, except for enhanced production of few compounds after an LP in OE::*tafrq* (*Table 1*).

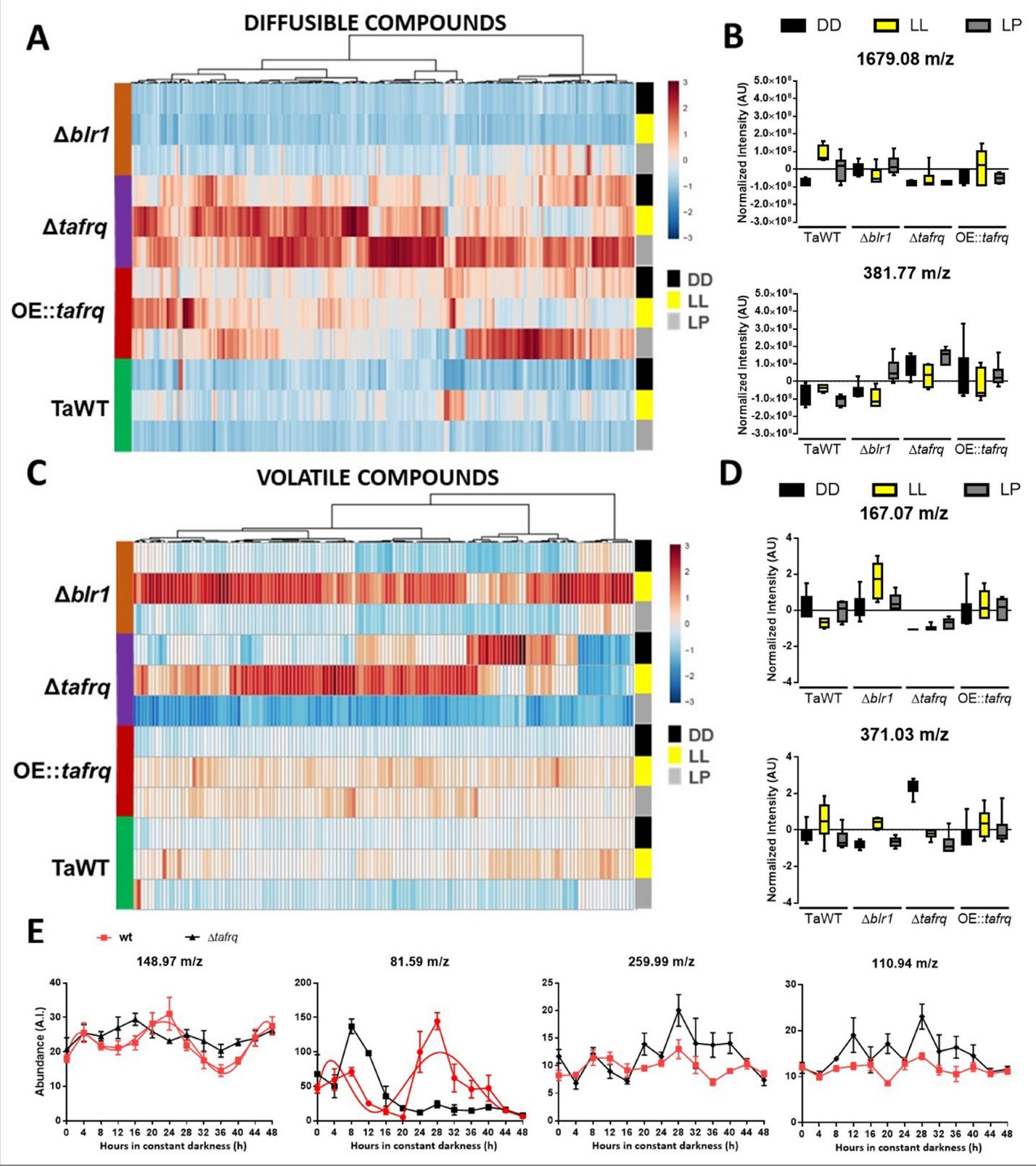

**Figure 6.** Secondary metabolism is regulated by light conditions and *T. atroviride* core clock components. A global overview of secondary metabolites produced by TaWT and clock mutant strains grown in LL, DD, and after a light pulse (LP) grown in potato dextrose agar (PDA) at 20°C. (**A**) Diffusible compounds are augmented in *Δtafrq*. (**B**) Normalized intensity values for two particular diffusible compounds are shown: 1679.08 m/z is light-induced in TaWT and light-repressed in *Δblr1* and *Δtafrq*; 381.77 m/z showed enhanced production in *Δtafrq* and OE::*tafrq* in all culture conditions. (**C**) Volatile

*Figure 6 continued on next page*

*Figure 6 continued*

organic compounds (VOCs) are induced by light in Δblr1 and Δtafrq. (**D**) Normalized intensity values for two VOCs: 6-PP (167.07 m/z) and 371.03 m/z. (**E**) For circadian VOCs production, TaWT and Δtafrq strains were grown in GYEC+ peas at 20°C during 3 days in LD 12:12 hr cycles and transferred to DD for circadian metabolic profiles. Samples were taken every 4 hr during 48 hr after the transition to DD. Circadian levels of ions 148.97 m/z and 81.59 m/z in TaWT, but arrhythmic in Δtafrq. Compounds 255.99 m/z and 110.94 m/z oscillate in neither strain.

The online version of this article includes the following figure supplement(s) for figure 6:

**Figure supplement 1.** Circadian production of volatile organic compounds (VOCs) in *T. atroviride*.

In TaWT, few VOCs depicted differences between light conditions, being one of them the well-known antifungal volatile compound 6-pentyl-α-pyrone (6-PP) (167.07 m/z). This compound showed to be induced in DD compared to LL, suggesting an inhibitory effect of light in 6-PP production (*Figure 6D*). In Δblr1, 6-PP production showed a 5.8-fold induction in LL, but no significant differences were observed in DD and LP compared to TaWT. A possible explanation is that BLR-1 could act as a repressor of 6-PP production in LL (which is consistent with the light repression of this molecule observed in TaWT and the augmented production in Δblr1 under LL); nevertheless, we cannot rule out that other photoreceptors could be involved in potentiating 6-PP production under LL in Δblr1 (*Figure 6D*). Interestingly, a dramatic reduction of 6-PP production was observed in Δtafrq in DD, LL, and LP (61, 10, and 3 negative fold change, respectively) (*Figure 6D*). These results clearly show that the profile of volatile compounds is also broadly altered in Δtafrq. The differences found in VOCs, particularly 6-PP production, correlate with (but cannot fully explain) the different biocontrol effectiveness of the tested strains in LL and DD conditions.

Finally, we performed a circadian metabolic profile measuring VOCs every 4 hr during 48 hr in DD conditions for TaWT and Δtafrq strains in GYEC+ peas plates. Rhythmicity of VOC levels was calculated using BioDare (with FFT-NLS and BD2eLTK cycle algorithms with a p-value<0.05), observing that 17 compounds exhibited a circadian profile with a calculated period between 24 and 26 hr in TaWT (*Supplementary file 4*). In Δtafrq, as expected, most compounds profiles (13 out of the 17) lost rhythmicity (see *Supplementary file 4*) and, importantly, the remaining four compounds (63.83 m/z, 90.83 m/z, 388.87 m/z, and 389.77 m/z) exhibited weak oscillatory behavior (e.g., reduced amplitude and altered phase) as determined after manual examination of compounds dynamics (*Figure 6—figure supplement 1*, *Supplementary file 4*). The circadian abundance of two ions (148.97 and 81.59 m/z) and two arrhythmic ions (259.99 and 110.94 m/z) are shown for comparison in *Figure 6E*. In total, these results suggest that the production of some VOCs in *T. atroviride* is under circadian control.

**Table 1.** Induced and repressed diffusible and volatile organic compounds (VOCs) in each mutant strain compared to TaWT in constant darkness (DD), constant light (LL), and light pulse (LP).

| | Diffusibles* | | | | | | VOCs† | | | | | |
|---|---|---|---|---|---|---|---|---|---|---|---|---|
| | DD | | LL | | LP | | DD | | LL | | LP | |
| | I | R | I | R | I | R | I | R | I | R | I | R |
| Δblr1/TaWT | 26 | 3 | - | 129 | 66 | - | - | 5 | 17 | - | - | - |
| Δtafrq/TaWT | 67 | - | 137 | 2 | 218 | - | 23 | 13 | 16 | 2 | - | 10 |
| OE::tafrq/TaWT | 34 | - | 38 | - | 128 | - | - | - | - | - | 10 | - |

Data can be found in *Table 1—source data 1*.

*Diffusible compounds with at least 2.0-fold change.
†Volatile organic compounds with at least 1.5-fold change.
I=induced, R=repressed.

The online version of this article includes the following source data for table 1:

**Source data 1.** Fold changes of each comparison pair (Δblr1/TaWT, Δtafrq/TaWT and OE::tafrq/TaWT) can be found in a single individual sheet in the provided .xlsx file.

## Discussion

While putative core clock components encoding genes can be found in fungi distributed between several orders (*Montenegro-Montero et al., 2015*; *Salichos and Rokas, 2010*), phenotypic description of fungal circadian phenomena has been less common, and the molecular characterization of those rhythms has been even scarcer. One of the few examples (besides *N. crassa*) is the characterization of the *B. cinerea* core clock and its effect on fungal virulence (*Hevia et al., 2015*). In our efforts to further understand the role of circadian clocks in fungal–fungal interactions, herein we provide evidence of a functional clock in the mycoparasitic fungus *T. atroviride*. TaFRQ$^{LUC}$ reporter strains revealed circadian oscillations with a free-running period of ~26 hr, which could be entrained by LD cycles and reset by light and temperature pulses, consistent with what has been described in *Neurospora* and other organisms for bona fide circadian systems (*Liu et al., 1998*). These oscillations showed temperature compensation between 20 and 25°C, where, notably, the best rhythms in terms of amplitude were observed at 22°C, but they were lost at 28°C. While such a narrow range of temperatures allowing clear visualization of these molecular rhythms may be puzzling, it might also be associated with the environment in which *Trichoderma* is usually found – soil – in which temperature shifts are overall buffered (*Rodriguez-Romero et al., 2010*).

In the attempt to evaluate clock-controlled genes (*ccg*), we did not observe oscillations of additional luciferase transcriptional reporters of putative *T. atroviride* ccg promoters. This could be due to (i) the lack of circadian transcription of the assayed ccgs, (ii) the selected promoters fail to capture elements necessary for their circadian control, or (iii) the assay conditions were not appropriate to visualize their rhythmic expression (see below). Future analysis of the transcript levels of several ccgs will be performed under different culture conditions to confirm whether their expression is actually circadian in particular settings.

Interestingly, we found evidence of strong functional circadian conservation between *Neurospora* and *Trichoderma*. Firstly, conservation of clock *cis*-regulatory motifs, as the *c-box* sequence from the former is recognized in the latter fungus. As the DNA binding domains of WC-1 and BLR1 share a 97.2% aa identity, we hypothesize that the BRL1/BLR2 complex can recognize the $_{Nc}$*c-box* element driving circadian oscillations and that a similar *c-box sequence* could be found in the *tafrq* promoter. Indeed, the high conservation of WC-1 and BLR1 DNA binding domains (*Casas-Flores et al., 2004*; *Cervantes-Badillo et al., 2013*; *Froehlich et al., 2002*) indicates that they are expected to recognize a similar motif considering general rules of eukaryotic transcription factor sequence specificity (*Weirauch et al., 2014*). Two putative light response elements in the *tafrq*'s promoter with a putative GATA box consensus sequence at proximal and distal locations (−355 to −319 and −1180 to −1171 from the TSS, respectively) have been postulated (*Cervantes-Badillo et al., 2013*), of which one of them could be acting as a putative $_{Ta}$*cbox* sequence.

We were able to generate ~26 hr rhythms in a *Neurospora* Δ*frq* strain complemented with the *tafrq* coding region, under the control of the *Neurospora* endogenous regulatory sequences. Comparison of the *tafrq* and nc*frq* sequences informs a 54 and 50% of identity at the DNA and protein levels, respectively (using Clustal Omega multiple sequence alignment), which provides sufficient functional conservation to allow rescuing of the core clock function in a *N. crassa* arrhythmic strain. Most importantly, the fact that *tafrq* complements the function of the endogenous *frq* in *Neurospora* strongly validates the role of TaFRQ as a core clock circadian component. While we did not conduct some classic experiments such as the ones designed by *Aronson et al., 1994*, we present compelling evidence of the existence of such a functional circadian TTFL. Thus, in addition to the complementation itself, it is the fact that TaFRQ is indeed acting as a negative element in *Neurospora*, which can be clearly inferred by the overall decrease of WCC activity (as seen by the reduction in luciferase expression when compared to the recipient strain). Importantly, as we restored the core clock in Δ*frq*, the resulting oscillations reflected the endogenous ~26 hr period originally observed in *T. atroviride*, instead of the *Neurospora* WT rhythm of ~22 hr, implying that period determinants are encoded in the TaFRQ protein sequence and that they are 'read' by the rest of the circadian machinery in a proper manner, despite the phylogenetic distance of both genera. Thus, this not only further validates TaFRQ as a bona fide core clock component, but it also highlights how its sequence determinants modulate period. Additionally, we also confirmed that TaFRQ is phosphorylated and degraded as seen over a 12 hr time course, after a light to dark transfer, in accordance to *Neurospora* FRQ behavior. Recent studies have shown that the quality of FRQ regarding phosphorylation status rather

than its stability or half-life is important for period length (*Larrondo et al., 2015*). *Baker et al., 2009* partially characterized FRQ's phosphocode in *N. crassa,* of which some mutations in these sites were experimentally shown to affect clock properties. In TaFRQ, 82.6% (19/23) of such sites are preserved, with the divergent ones all residing in the C-terminal portion of the protein, after the PEST-2 domain (*Figure 2—figure supplement 3*). The high conservation of known period-affecting phosphorylation sites could partially explain the ability of TaFRQ to function, whereas differential phosphorylation of TaFRQ compared to NcFRQ could potentially explain the longer period. Further analysis of TaFRQ characteristics and phosphorylation sites will shed additional light on the commonalities and peculiarities of the circadian clocks in these two fungi.

One of our study's striking findings was the strong influence of culture media composition in allowing the visualization of molecular circadian oscillations. Indeed, rhythms in TaFRQ$^{LUC}$, or in $_{Nc}$c-box-luc levels, were only observed in GYEC media and then further improved when plant-derived material was included. While the impact of nutrient availability in certain aspects of circadian rhythms has been observed from mammals to *N. crassa* (*Bae et al., 2019*; *Oosterman et al., 2015*), our results indicate an even bigger dependence on culture conditions in allowing molecular visualization of the *Trichoderma* clock. Thus, although the quality and other aspects of *frq* and clock-controlled genes can vary, depending on media characteristics in *Neurospora* (*Díaz and Larrondo, 2020*; *Dunlap and Loros, 2017*; *Hurley et al., 2014*; *Larrondo et al., 2015*; *Olivares-Yañez et al., 2016*) we were surprised to observe such a dramatic effect in *Trichoderma*: that is, an almost nutritional-conditional rhythmicity dependent on media-derived cues. Besides the puzzling and yet uncharacterized mechanism underlying this nutritional conditionality, we believe that this may serve as a cautionary tale for other attempts to characterize circadian rhythms in different fungi. If we had limited our analysis to a small set of media or higher temperatures, we would have been prone to conclude that there were no rhythms in *Trichoderma* TaFRQ levels. Yet, hitting the right conditions allowed us to confirm their existence and therefore the presence of a circadian oscillations in this organism. It is also provoking and an exciting lesson to consider that the addition of plant-derived material in a fungus that naturally interacts with plants has such a strong effect on visualizing these rhythms (see below).

In *T. atroviride*, the use of different carbon sources in culture media allows, in some cases, observation of photostimulation of growth and conidiation in the dark, processes in which BLR1 plays a critical role in carbon-source selectivity (*Friedl et al., 2008a*; *Friedl et al., 2008b*). Also, C:N ratios are relevant environmental factors influencing conidiation in *Trichoderma* (*Steyaert et al., 2010b*). Additionally, organic nitrogen sources in culture media have been described as important for fungal mycoparasitic behavior (*Barnett, 1963*). Also, nitrogen sources are known to affect growth in *Trichoderma* species (*Danielson and Davey, 1973*; *Nolan and Nolan, 1972*). Due to the importance of nutrients in such processes, we hypothesize that the high amounts of organic nitrogen found in GYEC media (derived from casamino acids and yeast extract, absent in the other tested media) to some extent may mimic the composition of *Trichoderma*'s natural niche, in which different nitrogen sources derived from its preys are available.

Moreover, the addition of peas could better resemble the nutrients or chemical features found in *Trichoderma*'s interaction with plants, including different sugars (*Druzhinina et al., 2011*; *Steyaert et al., 2010b*). The latter also raises fundamental questions about the importance of plant-derived signals in *Trichoderma*'s clock robustness and its role in the outcome of plant–fungal interactions. A complex temporal communication could be occurring during their interaction in the soil, affecting clock synchronization in *Trichoderma*. Indeed, the plant circadian clock could alter the composition and function of the rhizosphere *microbiome* as it has been observed in the rhizosphere of WT and long/short period mutants of *A. thaliana* (*Hubbard et al., 2018*). Rhythmically structured metabolic exchanges between the organisms in the community in collective benefit could be occurring (*Sartor et al., 2019*), although it still remains to be studied in this context.

These findings reinforce the importance of defined nutrient/temperature conditions to monitor the clockworks and help future fungal circadian studies visualize otherwise weak oscillations, like those observed in *Pyronema confluens* and *Ophiocordyceps kimflemingiae* (*de Bekker et al., 2017*; *Traeger and Nowrousian, 2015*). This could also help monitor *frq* molecular oscillations in fungi where previous attempts have failed, like in *Verticillium dahliae, Aureobasidium pullulans,* and *Magnaporthe oryzae* (*Cascant-Lopez et al., 2020*; *Deng et al., 2015*; *Franco et al., 2017*).

While *frq* in *N. crassa* does not have a described role outside the pacemaker (**Aronson et al., 1994**), in *B. cinerea*, it exhibits extra-circadian roles associated with developmental programs (**Hevia et al., 2015**). In *T. atroviride,* we observed that the core clock components play a role in development, secondary metabolism, and mycoparasitism. Thus, TaFRQ is critical for conidia formation after an LP and mechanical damage, showing a dramatic reduction of conidiation in the latter. In fungi, little is known about FRQ function besides *N. crassa* and *B. cinerea*. A recent study of *frq* in the phytopathogen *V. dahliae* found no significant phenotypic differences in the deletion strain regarding sporulation and microsclerotia formation in DD or LD (**Cascant-Lopez et al., 2020**). Similarly, in the entomopathogenic fungi *Metarhizium robertsii*, FRQ is necessary for ring formation under LD 12:12, although neither the WT nor the mutant strain displayed such rings under DD conditions. Besides this, *M. robertsii* FRQ is dispensable in almost all evaluated phenotypes (growth and conidiation in different stress conditions), including virulence against *Galleria mellonella* (**Peng et al., 2022**). Meanwhile, studies in *Beauveria bassiana* (in which two *frq* homologs exists) and *M. oryzae* strain *Guy11* have shown that their *frq* knockouts exhibit diminished conidiation and virulence against *G. mellonella* (**Tong et al., 2021**) and rice seedlings (**Shi et al., 2019**), respectively. These findings pose new questions regarding the role in developmental decisions that FRQ may have acquired (or lost) across the fungal kingdom, raising questions about the extra-circadian role of clock negative elements in other organisms, including animals.

We found altered developmental responses and enhanced production of diffusible and volatile compounds in *Δtafrq*, suggesting a general repressor regulatory role of TaFRQ over secondary metabolism. Interestingly, the antifungal compound 6-PP, a key feature of *Trichoderma* species, was dramatically reduced in *Δtafrq* even in LL. The clock function is expected to be abolished in LL, suggesting, therefore, a potential TaFRQ extra-circadian activator role in 6-PP production.

SMs have been related to developmental programs in fungi and could have a role in response to damage in *T. atroviride* (**Atriztán-Hernández et al., 2019**; **Schmoll et al., 2010**; **Tisch and Schmoll, 2010**). Reports show that after injury and fungivory a group of VOCs, including 6-PP, increase in TaWT but not in the MAPK mutant *Δtmk3* (**Atriztán-Hernández et al., 2019**), which shows diminished conidiation after an injury as observed in the *Δtafrq* mutant (**Medina-Castellanos et al., 2014**). Additionally, it was previously suggested that oxylipin could be involved in signaling after damage (**Medina-Castellanos et al., 2014**). We propose that the reduction of injury-induced conidiation in *Δtafrq* could be related to its dysregulated diffusible and VOC production profiles, in which further studies identifying those compounds could shed light on their role in injury-induced conidiation. On the contrary, *Δblr1* showed repressed production of several compounds (diffusible and VOCs), suggesting a role of light perception in secondary metabolism regulation. This is consistent with the light-induced production of several compounds in TaWT.

Interestingly, *Δblr1* produced higher amounts of 6-PP in DD and even in LL, whereas in TaWT, 6-PP production is inhibited by light. This suggests a repressor role of light and BLR1 in the production of 6-PP and other molecules, as seen for 1679.08 m/z, and that other photoreceptors could activate its production in the absence of BLR1 (**Garcia-Esquivel et al., 2016**). A light-repression effect on 6-PP production was also observed in *T. atroviride* P1 and IMI 206040 strains when 6-PP production was analyzed under LD and DD conditions (**Speckbacher et al., 2020a**; **Contreras-Cornejo et al., 2022**) and when grown in different light wavelengths (**Moreno-Ruiz et al., 2020**). We also observed differences in the SM profiles between LL and LP conditions, indicating that the duration of light stimuli differently activates/represses the production of several compounds. Whereas a short light exposure could be perceived as a strong environmental cue triggering the production of different metabolites, constant exposure to light could lead to a prolonged stress response marked by a different subset of metabolites. Light regulation of SMs has been reported in several fungi (**Tisch and Schmoll, 2010**) as the involvement of WC-1 homologs in *Cercospora zeae-maydis* and *Fusarium graminearum* as repressors of cercosporin and trichothecene production, respectively (**Kim et al., 2014**; **Kim et al., 2011**). Nevertheless, the involvement of TaFRQ in the production of SMs in *T. atroviride* (or FRQ of any other fungus) has not been reported to date.

In concordance with the circadian oscillations of TaFRQ^LUC, we observed rhythmic levels of volatile SMs in *T. atroviride* under constant conditions (DD) but not in *Δtafrq*, suggesting circadian control of the production of these volatile compounds. Our experimental approach showed that these molecules exhibited rhythmic levels. Nevertheless, it did not provide their molecular identity. We

expect future work to dive into their chemical identification and their specific role in *T. atroviride*'s lifestyle.

Overall, our results reinforce the importance of environmental perception (including time) in *Trichoderma*'s biology, particularly in the production of SMs involved in several processes from development to mycoparasitism. On the contrary, in *Neurospora* how these daily rhythms impact metabolite levels has not been described so far, despite ample efforts that have systematically shown the clock's influence on its transcriptome and proteome (*Hurley et al., 2014*; *Hurley et al., 2018*; *Sancar et al., 2015*), suggesting temporal compartmentalization of metabolism at large, with catabolic and anabolic reactions concentrated at dawn and dusk, respectively (*Hurley et al., 2016*). Thus, we present a proof of concept that a fungal circadian clock can also impact the production of SMs. These studies and future comprehensive analyses of circadian metabolic profiles in classic models such as *Neurospora* and other fungi will help to bridge the gap on how clock regulation impacts a fungus's daily life, from physiology to organismal interactions.

Our study also provides new insights into the mycoparasitic interaction between *T. atroviride* and *B. cinerea* and how its outcome is regulated by core clock components and LD conditions. *Δblr1* showed the strongest mycoparasitic behavior; in contrast, *Δtafrq* did not show significant differences to TaWT in most interactions, except a slightly weaker mycoparasitic behavior against *Δbcwcl1* and *Δbcfrq1* in LL. *Trichoderma* species' antagonistic capacity is correlated with their ability to produce 6-PP (*Reino et al., 2008*). The *Δblr1* strain has higher amounts of this molecule, whereas, in *Δtafrq*, its production is severely dampened, consistent with their different mycoparasitic abilities. *Trichoderma*'s antagonist capacity was reduced in LL and enhanced in darkness, which could be due to differential metabolite profiles found in LL and DD. Similar inhibitory effects of light in mycoparasitism in *T. atroviride* have been reported against *Fusarium oxysporum* (*Moreno-Ruiz et al., 2020*; *Speckbacher et al., 2020b*), along with differential VOC production upon illumination during confrontation assays. High amounts of 2-heptanone are produced during the confrontation against *F. oxysporum* in a light-dependent manner that negatively correlates with *Trichoderma* antagonism, which proposes that 2-heptanone production was a result of a stress response elicited by *F. oxysporum* in the light.

Finally, mycoparasitic interactions under a circadian paradigm revealed an enhanced overgrowth of TaWT and *Δtafrq* over B05.10 and *Δbcwcl1* when interactions started at dawn under LD cycles in contrast to a reduced overgrowth at dusk. We have previously observed differential interactions in the context of *B. cinerea* and *A. thaliana*, where fungal virulence was enhanced at dusk (*Hevia et al., 2015*). The differential effect on the *Trichoderma–Botrytis* dynamic was lost when *Δblr1* was used against all *B. cinerea* strains. A similar phenomenon was observed by *Hevia et al., 2015*, when *A. thaliana* was infected with *Δbcwcl1*, which did not show circadian behavior. Disruption of BLR1 abrogates both light and circadian responses, so the effect observed for *Δblr1* could also be due to a light-driven phenomenon (besides a circadian effect) because the absence of the blue-light perception overrides the differential effect observed in TaWT. Yet, it is noteworthy that *Δtafrq* yields result comparable to what is seen in TaWT, suggesting that – paradoxically – the observed time-of-the-day effect does not appear to strongly depend on the *T. atroviride*'s clock.

On the other hand, under LD cycles, the loss of BcWCL1 does not considerably affect these fungal dynamics, whereas, in *Δbcfrq1*, the differential outcome of the daily interaction disappears. Lack of BcFRQ1 implies that temporal perception in *B. cinerea* is absent, in concordance with the loss of the time-of-the-day observed phenotype, something that is not seen when *Trichoderma* lacks TaFRQ. This suggests that the *B. cinerea* clock is critical in the outcome of the interaction of tested organisms – both plants and fungi – suggesting a complex mechanism in which light and time perception interact, giving rise to the observed phenotypes. Even though we lack a precise molecular understanding underlying this phenomenon (and integrating the role of the different clock/light-perception components), this work constitutes the first description of an interplay of core and light regulation in a mycoparasitic interaction. In nature, organisms are usually exposed to light/dark transitions. Therefore, the differential outcome depending on the time (or the light/dark cycles) at which confrontations are established could impact biocontrol strategies in agriculture, optimizing the application of *Trichoderma* in crops.

## Materials and methods

### Fungal strains and culture conditions

*T. atroviride* IMI 206040, initially isolated from a plum tree of the Czar variety in an orchard in Urshult near Växjö in southern Sweden (*EFSA European Food Safety, 2015*), was used as wild-type strain (TaWT) and, along with *Δblr1*, described by *Casas-Flores et al., 2004*, were kindly provided by Dr. Alfredo Herrera-Estrella. Strains were maintained in potato dextrose agar (PDA) in constant light at 25°C. *B. cinerea* B05.10 Pers. Fr. [*Botryotinia fuckeliana* (de Bary) Whetzel] was initially isolated from *Vitis vinifera* (Germany) and was provided by the Tudzinsky laboratory (WWU, Germany). *B. cinerea* clock mutant strains *Δbcwcl1* and *Δbcfrq1* used in this study were described earlier (*Canessa et al., 2013*; *Hevia et al., 2015*). *B. cinerea* strains were cultivated in PDA at 20°C in LD 12:12 hr cycles. All fungal strains were grown in Percival incubators, equipped with cold white-light fluorescent tubes (light intensity up to 100 µM/m²s; wavelength 400–720 nm). The incubators were housed inside a darkroom equipped with safety-red lights to allow proper circadian experiments.

### Generation of *Δtafrq*, *Δtafrq*C, and OE::*tafrq* mutant strains

The *tafrq* ORF was replaced by the hygromycin resistance cassette (*hph),* as described (*Canessa et al., 2013*; *Figure 3—figure supplement 1*, *Supplementary file 5*). To generate the complemented *Δtafrq* strain (*ΔtafrqC*), we fused the *tafrq* ORF with a fleomycin resistance cassette (*bleoR*) and directed it back to its endogenous locus by homologous recombination (*Figure 3—figure supplement 3*, *Supplementary file 5*). To overexpress *tafrq*, we placed *tafrq* under control of the *T. atroviride* actin promoter (1210 bp) with *hph* as a selection marker. All constructions were assembled using yeast chromosomal recombination (*Oldenburg et al., 1997*). The OE::*tafrq* construct was integrated by homologous recombination into the intergenic *blu17* region, previously demonstrated not to affect mycoparasitism behavior (*Supplementary file 6*; *Balcazar-Lopez et al., 2016*).

### Generation of luciferase reporter strains

The TaFRQ^LUC strains were designed and generated, as previously reported by *Larrondo et al., 2012*. Briefly, the *tafrq* gene was fused by homologous recombination at its endogenous locus to an *N. crassa* codon-optimized luciferase gene (*luc*), and *hph* used as a selection marker (*Figure 1—figure supplement 1*, *Supplementary file 7*). On the other hand, the transcriptional reporter strain, termed $_{nc}$*c-box-luc,* was designed as reported by *Larrondo et al., 2015*. We cloned a 505 bp sequence containing the *c-box* sequence from the *N. crassa frq promoter* to control *luc* expression, and the construct was integrated into the intergenic region of *blu17* (*Supplementary file 8*; *Balcazar-Lopez et al., 2016*). To develop transcriptional reporters of hypothetical clock-controlled genes (*ccg*) in *T. atroviride,* we used a promoter region of different genes as follows: 768 bp from TRIATDRAF_77441, homologous to *ccg-9* (trehalose synthase in *N. crassa*), 1159 bp from TRIATDRAF_297741, homologous to glyceraldehyde 3-phosphate dehydrogenase of *N. crassa*, 485 bp from TRIATDRAF_298583 homologous to superoxide dismutase of *N. crassa* and 1000 bp from TRIATDRAFT_291013 homologous to *con-10*. The promoters were used to control *luc* expression, and the constructs were integrated into the intergenic *blu17* region (*Supplementary file 8*; *Balcazar-Lopez et al., 2016*).

### Heterologous expression of *tafrq* in *N. crassa Δfrq* strain

A construct containing a 5′ recombination flank (1000 bp), the native 5′UTR from *N. crassa frq*, the *tafrq* ORF followed by a V5-His6 tag, the native 3′UTR from *frq*, the *bar* cassette for resistance selection (*Larrondo et al., 2009*), and a 3′ recombination flank (1000 bp) was constructed using yeast recombination cloning (*Oldenburg et al., 1997*). The resulting construct was used to transform a *N. crassa Δfrq* strain containing a *his-3::frqcbox-luc* reporter (*Figure 2—figure supplement 2*). To obtain homokaryotic isolates, strains complemented with *tafrq* were crossed to a WT strain containing *Neurospora frq* (not associated with any marker or tag) and lacking a reporter. Ignite-resistant, luc+ progeny was selected out of the cross, and the targeted integration of *tafrq* (and absence of *Neurospora frq*) was confirmed by PCR (*Supplementary file 9*). *Δncfrq:tafrq* strains were assessed for conidial banding using race tubes grown in two different conditions, DD conditions for 8 days and LD (12:12 hr) for 5 days followed by DD for 3 days, alongside positive and negative controls, as described in *Baker et al., 2012*.

## Transformation of *T. atroviride*

PEG-mediated protoplast transformation was performed as described by *Casas-Flores et al., 2004* and *Herrera-Estrella et al., 1990*. All transformants were subjected to 3–5 rounds of single-spore isolation, and gene replacement and integration events were verified using Southern blot using the DIG Easy Hyb Hybridization solution (Roche) following the manufacturer's instructions. The *hph* gene was used as a probe (*Figure 1—figure supplement 1*, *Figure 3—figure supplement 1*).

## Assessment of luminescence in vivo

For evaluating rhythmic levels of bioluminescence produced by the different luciferase reporter strains in *T. atroviride*, spores from 7-day-old colonies were resuspended in sterile water, filtrated using sterile glass wool, and kept for 48 hr at 4°C. Then, 30 µL of spore suspension were inoculated in individual wells of 96-well plates containing different culture media (*Supplementary file 1*). Luciferin was added to each media to a final concentration of 2.5 mM. The reporter strains received LD 12:12 entrainment at 20, 22, 25, and 28°C. Luminescence traces were captured using a PIXIS 1024B CCD camera (Princeton Instruments) under constant darkness (DD) conditions at the indicated temperatures. For LL (constant light) resetting, strains were grown 72 hr in continuous light at 25°C before transferring to DD for bioluminescence measurements. For the 4°C pulses, strains were incubated for 48 hr in LL at 25°C and then moved to 4°C for 6 hr in complete darkness before measurements. For the LP experiments, strains were incubated for 48 hr in LL, transferred to DD for 24 hr, and then received a 20 min LP at 25°C, and transferred back to DD. Bioluminescence was acquired using the WinView software (version 2.5.23.0) and subsequently analyzed using ImageJ employing a custom-made macro. As indicated therein, we present raw or normalized data (align to mean) in different graphs. Circadian parameters, such as period and phase, were calculated using FFT-NLLS, which fits a sine wave to the data to estimate period, phase, and amplitude using the BioDare online platform (*Zielinski et al., 2014*).

The different tested culture media are indicated in *Supplementary file 1*. In particular to prepare GYEC+ peas (0.5% W/V), we extensively grind frozen commercial peas in a coffee grinder and saved it as a frozen powder.

For the evaluation of *N. crassa* luciferase reporter strains, spores from 5-day-old slant cultures containing minimal media were resuspended in sterile water. Then, 15 µL of spore suspension were inoculated in individual wells of 96-well plates containing LNN-CCD media (*Larrondo et al., 2015*) supplemented with quinic acid (QA, 0.01 M). Luciferin was added to each media to a final concentration of 25 µM. The reporter strains received LD 12:12 entrainment at 25°C for 72 hr prior to CCD monitoring.

Luciferase signals were acquired with a PIXIS 1024B camera (Princeton Instruments) and quantified with a customized macro developed for the ImageJ software; luminescent signals were recorded for 5 min, three times per hour for 5 days. Circadian parameters, such as period and phase, were calculated using fast Fourier transform nonlinear least squaresFFT-NLS, which fits a sine wave to the data to estimate period, phase, and amplitude using the BioDare 2.0 online platform (biodare2.ed.ac.uk; *Zielinski et al., 2014*). For *N. crassa*, linear detrending was performed before the period estimation.

## RNA extraction and reverse transcription

For overexpression analysis of *tafrq* in *T. atroviride*, strains were grown in PDA cellophane-covered plates for 48 hr in LL or DD at 25°C. Approximately 10 mg of tissue was collected from each plate and snap-frozen in liquid nitrogen. Frozen mycelia were ground to powder and total RNA was extracted using TRIzol. Total RNA quantity and quality were determined using a NanoDrop spectrophotometer. Three independent biological replicates for each condition were analyzed.

Prior to cDNA synthesis, 3 µg of total RNA was treated with RQ1 RNase-free DNase and the absence of contaminating genomic DNA in the RNA samples was checked with real-time PCR (qPCR), with RNA as template (i.e., –RT control; data not shown). Reverse transcription was performed on 1 µg of total RNA using SuperScript III First-Strand Synthesis SuperMix for RT-qPCR.

Assay efficiency was assessed using serial dilutions (1/10) of a qPCR product (one product for each set of primers). Each dilution was used as a DNA template, and a standard qPCR reaction was performed. Primer efficiency was determined using the qPCR Software CFX Maestro (Bio-Rad). Specificity was evaluated by both melt curve and agarose gel analyses. All qPCR reactions were run in the CFX Real-Time PCR – Bio-Rad utilizing SensiMix SYBR Hi-ROX kit and primers for *tafrq* as well as normalizing genes (*Supplementary file 10*).

## Protein extracts and immunoblots

To detect TaFRQ$^{LUC}$ oscillations at protein levels, TaFRQ$^{LUC}$ reporter strains were subjected to a time-course experiment in GYEC+ peas culture media, for which samples were taken every 4 hr for 52 hr, following a time-course protocol similar to the ones in *Neurospora* (*Larrondo et al., 2015*), except that *T. atroviride* cultures were kept in DD, and reset by a strong LP to mimic an LL to DD transition. Proteins were extracted as described by *Kilani et al., 2020*. Protein concentration was determined by the Bradford method, using the Bio-Rad Protein Assay (Bio-Rad Laboratories, Inc, CA). For immunoblot analyses, total protein extracts (50 µg of protein per lane) were separated in 4–15% Mini-PROTEAN TGX Precast gels (Bio-Rad #456-1086) and transferred to PVDF membranes (Bio-Rad) using the Trans-Blot Turbo RTA Transfer Kit (Bio-Rad #170-4272). The membranes were blocked overnight in phosphate-buffered saline solution (PBS-T: 137 mM NaCl, 2.7 mM KCl, 10 mM Na$_2$HPO$_4$, 2 mM KH$_2$PO$_4$, and 0.1% Tween-20) with 5% non-fat dried milk at 4°C. To detect FRQ$^{LUC}$ protein, a luciferase antibody C-12 (sc-74548, Santa Cruz Biotechnology Inc, TX) was used at a 1:250 dilution and as secondary antibody Goat Anti-Mouse IgG(H+L)-HRP conjugate at a 1:5000 dilution.

For a mini time course of TaFRQ expression in *N. crassa*, 30 µL of resuspended spores from 5-day-old slant were inoculated in plates containing a liquid medium comprised of 1× Vogel's salts, 2% sucrose, 0.5% arginine, and 50 ng/mL biotin for 2 days at 25°C in constant light conditions. From these pre-inoculates, tissue pads of 5 mm were cut and inoculated on plates with solid media using cellophane-covered Petri dishes with 25 mL LNN-CDD media supplemented with 0.01 M QA (*Larrondo et al., 2015*). For each short time course, plates were inoculated in triplicate per strain and incubated at 25°C in constant light. After 48 hr, the first cultures were harvested in light (time = 0 hr) while the others were transferred to dark conditions (25°C) and harvested every 2 hr under a safe red light. Samples were ground in liquid nitrogen and protein extracted as described above. As positive and negative controls, plates of *N. crassa* expressing FRQ$^{V5}$ (XC2116-10) and the *Δfrq* recipient (XC1783-4a) strains, respectively, were harvested after 48 hr in light.

For Western blot analysis, between 40 and 50 µg of total protein was loaded per lane after determination of concentration by Bradford assay (*Bradford, 1976*). Anti-V5 antibody (Invitrogen) was used at 1:5000 and a as secondary one we utilized Anti-Mouse IgG(H+L)-HRP (Thermo Fisher). SuperSignal West Pico ECL (Pierce Chemical, Rockford, IL) was used for signal development. For dephosphorylation reactions, lambda protein phosphatase (New England Biolabs, Cat# P0753L) was used according to the manufacturer's instructions.

## Confrontation assay between *B. cinerea* and *T. atroviride*

To evaluate the role of circadian regulation on *B. cinerea* and *T. atroviride* interactions, confrontations assays were conducted as a modified version of *Bell, 1982*. Overgrowth was calculated as the percentage of the *B. cinerea* colony that was covered by *Trichoderma*. TaWT and clock mutant strains of both fungi were grown in PDA at 20°C for *B. cinerea* and PDA and GYEC at 25°C for *T. atroviride*, during 3 days either in constant light (LL), darkness (DD) (*Figure 4A*), or in opposite LD cycles (LD AM and LD PM) (*Figure 5A*). Mycelial plugs of each strain were inoculated on opposite sides of 90 mm × 15 mm Petri dishes, 1 cm from the edge, and kept at 20°C in LL, DD, and LD conditions (as indicated in each case). Digital images were acquired after 7 days of interaction. The analyses of the photos were performed using the ImageJ software, employing an external calibration scale.

## Conidiation response assays

For light-induced conidiation, we cultivated *Trichoderma* strains in PDA at 25°C in LL for 7 days and collected and quantified conidia at 3 and 7 days of growth. For LP conidiation assays, strains were grown in PDA at 25°C in DD for 36 hr. White LPs of 1 or 60 min were given, and cultures were maintained in DD for an additional 48 hr, after which conidia were quantified. For injury-induced conidiation, strains were grown in PDA at 25°C in DD for 32–36 hr and damaged with a scalpel in DD using a red safety light and incubated for an additional 48 hr in the dark. Subsequently, as in previous cases, conidia were collected in sterile water and quantified by direct counting in a Neubauer chamber. For statistical analyses, ANOVA was used, and significance was determined using Tukey's test with a threshold of p<0.05.

## Secondary metabolic fingerprinting

The effect of core clock components and light in *Trichoderma* SM profiles was determined by growing strains for 48 hr in PDA at 25°C in LL or DD. A group of dark-growing strains received a 4 hr LP. TaWT and *Δtafrq* strains were grown in GYEC with 0.5% peas during 3 days in LD 12:12 hr cycles and transferred to DD for circadian metabolic profiles. Samples were taken every 4 hr during 48 hr after the transition to DD.

Diffusible compounds analyses were carried out using direct liquid injection mass spectrometry (DLI-MS). Mycelia of the strains grown in PDA in LL, DD, and LP conditions, were harvested and extracted with 1 mL of a mix of high-performance liquid chromatography (HPLC)-grade ethyl acetate, methanol, and dichloromethane 3:2:1 with 1% formic acid. The samples were sonicated for 20 min at room temperature at 40 kHz and centrifuged at 15,000 rpm for 20 min. The supernatant was transferred to a clean Eppendorf tube, and the solvent was evaporated at room temperature. Extracts were weighed and resuspended in methanol to 0.05 mg/mL. A 1/10 dilution was injected in the DLI-MS in positive ESI mode using an LCQ Fleet ion trap mass spectrometer. The spectra were acquired in the positive mode in an m/z range of 500–2000. For VOC analysis, the LTP-MS device was used as described by *Martínez-Jarquín et al., 2016* and *Martínez-Jarquín and Winkler, 2013*. The Petri dish was placed under the plasma. The spectra were acquired in the positive mode in an m/z range of 50–500. In both analyses (DLI and LTP MS), six biological replicates were used for each condition.

## Data analysis

The mass spectra raw data were converted to mzML open format with ProteoWizard (*Kessner et al., 2008*) and analyzed in the R statistical programming language using the MALDIquant package (*Gibb and Strimmer, 2012*). All spectra were aligned and corrected for compiling a data matrix. For LL, DD, and LP culture conditions, data were filtered using interquartile range (IQR) and scaled using autoscaling (mean-centered and divided by each variable's standard deviation). Significant differences in mean abundance among metabolites in different samples were assessed with ANOVA, and significance was determined using Tukey's test with a threshold of $p < 0.05$. A heatmap correlation matrix was built to visualize metabolic variations in all analyzed strains and experimental conditions with statistically significant metabolites (ANOVA). The Euclidean distance and Ward algorithm were used for clustering in the MetaboAnalyst platform (*Pang et al., 2020*). Fold change and *t*-test were calculated in pairs using the mean value of raw data for each comparison. A 2-fold change cutoff was established for diffusible and 1.5-fold change for volatile compounds.

For circadian VOC studies, a Bayesian analysis was executed using all 10 micro-scans taken as individual scans to identify ions related to the sample or noise (ambient noise and for GYEC+ 0.5% peas medium). The Bayesian analysis shows the percentage of the ion related to the sample: ions with 0% were eliminated. Adjustment of intensities was made on the spectra matrix made with means of the micro-scans. Determination of period and rhythmicity was carried out in the BioDare platform (*Zielinski et al., 2014*) using FFT-NLLS and BD2eJTK with $p < 0.05$.

## Acknowledgements

We thank Orlando Camargo, Tzitziki González, and Abigail Moreno for their invaluable help. This research was funded by ANID – Millennium Science Initiative Program – Millennium Institute for Integrative Biology (iBio ICN17_022), ANID/FONDECYT 1211715, ANID/FONDECYT-Postdoctoral 3180328 (to AS), ANID/FONDECYT-Postdoctoral 3190628 (to CO-Y), and the International Research Scholar program of the Howard Hughes Medical Institute. We also acknowledge Fundación Ciencia & Vida for providing infrastructure and laboratory space for experiments.

## Additional information

#### Competing interests
Luis F Larrondo: Reviewing editor, *eLife*. The other authors declare that no competing interests exist.

## Funding

| Funder | Grant reference number | Author |
| --- | --- | --- |
| Agencia Nacional de Investigación y Desarrollo | FONDECYT Regular 1211715 | Luis F Larrondo |
| Agencia Nacional de Investigación y Desarrollo | FONDECYT Postdoc 3180328 | Aldo Seguel-Avello |
| Agencia Nacional de Investigación y Desarrollo | FONDECYT Postdoc 3190628 | Consuelo Olivares-Yánez |
| Howard Hughes Medical Institute | the International Research Scholar program | Luis F Larrondo |
| Agencia Nacional de Investigación y Desarrollo | Millennium Science Initiative Program - Millennium Institute for Integrative Biology (iBio ICN17_022 | Paulo Canessa Luis F Larrondo |

The funders had no role in study design, data collection and interpretation, or the decision to submit the work for publication.

## Author contributions

Marlene Henríquez-Urrutia, Conceptualization, Data curation, Validation, Investigation, Visualization, Methodology, Writing - original draft; Rebecca Spanner, Aldo Seguel-Avello, Hector Guillén-Alonso, Investigation, Methodology, Writing – review and editing; Consuelo Olivares-Yánez, Data curation, Investigation, Methodology, Writing – review and editing; Rodrigo Pérez-Lara, Investigation, Methodology; Robert Winkler, Conceptualization, Resources, Data curation, Supervision, Validation, Methodology, Writing – review and editing; Alfredo Herrera-Estrella, Conceptualization, Resources, Data curation, Formal analysis, Supervision, Visualization, Writing – review and editing; Paulo Canessa, Conceptualization, Resources, Data curation, Supervision, Funding acquisition, Visualization, Writing - original draft, Writing – review and editing; Luis F Larrondo, Conceptualization, Resources, Formal analysis, Supervision, Funding acquisition, Writing - original draft, Project administration, Writing – review and editing

## Author ORCIDs

Luis F Larrondo ⓘD http://orcid.org/0000-0002-8832-7109

## Decision letter and Author response

Decision letter https://doi.org/10.7554/eLife.71358.sa1
Author response https://doi.org/10.7554/eLife.71358.sa2

# Additional files

## Supplementary files

• Supplementary file 1. Culture media used for in vivo luminescence assessment.

• Supplementary file 2. Period analysis of transcriptional reporters in *T. atroviride* using BioDare platform.

• Supplementary file 3. Mean overgrowth area of *T. atroviride* over *B. cinerea* in all confrontation assays performed.

• Supplementary file 4. Circadian metabolites of *T. atroviride*. The period was estimated using FFT-NLSS, and rhythmicity was evaluated using BD2eJTK in the BioDare platform.

• Supplementary file 5. List of primers used for *tafrq* replacement cassette, complementation cassette, and diagnostics PCR. Fw: direct orientation; Rv: reverse orientation.

• Supplementary file 6. List of primers used for OE::*tafrq* insertion cassette. Fw: direct orientation; Rv: reverse orientation.

• Supplementary file 7. List of primers used for luciferase translational reporter (TaFRQ$^{LUC}$) insertion cassette. Fw: direct orientation; Rv: reverse orientation.

• Supplementary file 8. List of primers used for luciferase transcriptional reporter insertion cassette.

Fw: direct orientation; Rv: reverse orientation.

• Supplementary file 9. List of primers used for *frq::tafrqV5* replacement cassette in *N. crassa* and diagnostic PCRs. Fw: direct orientation; Rv: reverse orientation.

• Supplementary file 10. RT-qPCR primers used and setting conditions.

• Transparent reporting form

## Data availability

All data generated and analyzed during this study are included in the manuscript and supporting files. Source data files have been provided for Figures 1 and 2 and Table 1.

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
