## [Editor Report]

The article by Henríquez-Urrutia and colleagues aims to establish that the fungus *Trichoderma atroviride*, a major biocontrol agent, has a circadian clock and test if this clock plays a role in development, secondary metabolite production, and mycoparasitism. The study is an important contribution toward the development of a new system for understanding how circadian rhythms influence fungal–fungal interactions that rests on solid experimental evidence.

---

## [Decision Letter]

**Decision letter after peer review:**

Thank you for submitting your article "Circadian oscillations in *Trichoderma atroviride* and the role of core clock components in secondary metabolism, development, and mycoparasitism against the phytopathogen *Botrytis cinerea*" for consideration by *eLife*. Your article has been reviewed by 3 peer reviewers, and the evaluation has been overseen by a Reviewing Editor and Detlef Weigel as the Senior Editor. The following individual involved in review of your submission has agreed to reveal their identity: Kevin Fuller (Reviewer #2).

Essential revisions:

The reviewers have made several constructive suggestions that you should carefully consider and address in a point-by-point reply. In particular, please pay particular attention to the following revisions:

1) Please perform experiments that demonstrate entrainment (as suggested by Reviewer #1).

2) Please perform the interaction assays across a range of environmental conditions (nutritional, temperature) to determine if the contributions of the clock proteins (in either organism) change accordingly (as suggested by Reviewer #2).

3) Please re-complement your mutants (as suggested by Reviewer #3).

*Reviewer #1:*

The goal of this study was to establish that the fungus Trichoderma atroviride, and important biocontrol agent, has a circadian clock, and to test if this clock plays a role in development, secondary metabolite production, and mycoparasitism. The authors clearly establish rhythms in the organism that are dependent on TcFRQ, a homolog of the core clock component FRQ in *N. crassa*, and show that TcFRQ and BRL1, the homolog of the blue light photoreceptor and core clock component WC-1 in *N. crassa*, play differing roles in development, metabolite levels, and mycoparasitism in laboratory conditions. However, the authors do not fully establish that the rhythms are controlled by a circadian clock , nor that TcFRQ is a core clock component. While aspects of this work will be of interest to circadian and fungal biologists, the not all of the aims were met, and the data do not fully support the conclusions.

Strengths: The circadian clock in the fungus *N. crassa* serves as an important model for understanding the mechanisms that keep time in eukaryotes, and how this clock controls daily rhythms in gene expression. However, little is known about clocks in other fungi, despite conservation of core clock components found in *N. crassa*, including fungal pathogens and biocontrol agents like Trichoderma. Thus, the strength of this work is in that it goes a long way to establish a clock in T. atroviride that is directly, or indirectly involved in controlling development, metabolism, and mycoparasitism. Importantly, this work opens up the opportunity to determine how the clock impacts fungal-fungal interactions, with the potential of using this to improve biocontrol.

Weaknesses: There are several weaknesses. The first is that while the authors established conditions to demonstrate two canonical properties needed to establish a circadian clock, a free running rhythm that is close to 24 h and temperature compensation, the authors did not fully test one of the cardinal properties of a circadian clock, entrainment. Second, the authors do not establish that TcFRQ is a core component of the clock oscillator. Third, the authors made conclusions from a strain that is supposed to overexpress TcFRQ from an actin promoter but, there is no data demonstrating that TcFRQ is actually overexpressed. Fourth, the rationale for concluding that TcFRQ has an extra-clock role, as opposed to a clock role, in development and metabolism is not clear Finally, a red safety light was used in all circadian experiments in constant dark; however, there is no data to support that T. atroviride is not responsive to red light.

Figure 1 While these data support the existence of a circadian clock in this organism based on free running rhythms and temperature compensation, it is important to provide additional data to demonstrate that the rhythm is entrained, one of the 3 criteria established in the field.

a. To demonstrate circadian entrainment, as opposed to a direct response to the environmental cues, specifically light, they need to examine different length LD cycles to demonstrate that the clock entrains to the different cycles. In addition, to demonstrate that an acute light or temperature pulse is resetting the clock, as opposed to masking, a a phase response curve is needed to show advances and delays. It is not clear what phase the light pulse was given in Figure 1D.

b. In the Supplement to Figure 2, several genes are arrhythmic under conditions in which TAFRQLUC is rhythmic. This is an important control to show, for example, that there is not a rhythm in ATP levels that might be driving the rhythm in the TAFRQLUC reporter. I would like to see this mentioned earlier in relation to Figure 1.

c. It would be helpful to know circadian time in Figure 1, and of interest to know if the peak time is similar to *N. crassa* FRQ.

Figure 2 demonstrates that the cis-acting promoter element of *N. crassa* frq that bound by the positive element, the WCC, is necessary for frq mRNA rhythms and functioning of the feedback loop, is sufficient to drive rhythms in luciferase in T. atroviride in DD, and is responsive to light and temperature pulses. This experiment suggests that there is conservation of the c-box, and raises the possibility that TAFRQ functions similarly in the T. atroviride clock as a negative element; however, this experiment raises more questions than it answers, and therefore seems premature. There are several instances in the paper where TaFRQ is called a core clock component, but there are no data to support this claim.

a. Is there a similar sequence present in the promoter of TAFRQ? If so, what happens if this sequence is mutated?

b. What happens to the rhythm, levels, of TAFRQ in a Blr1 deletion or mutated strain?

c. Additional experiments are needed to show that Blr1 binds to this sequence and to fully describe TAFRQ role in the clock mechanism.

In Supplementary file 1, 2A, B and C, they examine if TaFRQ has a role in development, and this is supported by the data presented by comparing WT versus TaFRQ deletion strains. They show that conidia formation is delayed in the mutant compared to WT, and decreased following light treatment or injury. In addition, they also examined a construct aimed at overexpressing TaFRQ (OE::TaFRQ) using the actin promoter. No differences were observed between WT and OE::TaFRQ; however, it is difficult to make any conclusions here because there were no experiments demonstrating that TaFRQ was actually overexpressed in this strain.

Figure 3 examines the ability of T. atroviride to overgrow B. cinerea under different lighting conditions and in mutants as an indicator of mycoparasitic behavior. The key findings were that constant light inhibits overgrowth, and that mutations in blr1 in T. atroviride enhances overgrowth in constant darkness. However, it is not clear how these data and those that follow establish and extra-clock role for TaFRQ in these processes.

Figure 4 looks at dawn and dusk confrontations between T. atroviride and B. cinereai in LD cycles. They found that WT T. atroviride and TaFRQ deletion strains had overgrowth that was enhanced at dawn, whereas deletion of TaBLR1 did not show a time of day difference. However, when frq was deleted in B. cinerea, all T. atroviride strains overgrew. These data supported that homologs of *N. crassa* clock components in T. atroviride and B. cinereai play differing roles in time of day specific interactions, and this information could prove useful in biocontrol measures in the future.

Figure 5 examines if T. atroviride FRQ and BLR1 mutants alter secondary metabolism by examining diffusible and volatile compounds in DD and LL, and following a light pulse. They found several differences, but were unable to correlate these with the overgrowth experiments in Figures 3 and 4. A circadian metabolic profile demonstrated that some compounds were rhythmic in WT and arrhythmic in taFRQ deletion cells. These data provide evidence to support a role for a clock in controlling metabolism in T. atroviride.

Discussion: The paper lacks definitive experiments to show that taFRQ is a core clock component, and therefore, it is not clear how these data support a extra-circadian role for taFRQ. In addition, aspects of the discussion are very lengthy, especially the section describing how this work could foster looking for clocks in other fungi.

Methods:

a. It is critical to show that T. atroviride is not sensitive to red light given red safety lights were used in circadian experiments.

b. The constructs used to examine potential rhythms in known fungal ccgs used the promoter, but not the 3' ends of the genes. The authors may want to consider adding this in the future.

Supplemental Figures

1.4 In part A, the light pulse did not seem to improve the rhythm similar to what was shown in Figure 1. What the light pulse given at the same time of day?

1.5. When was the light pulse given? These data should be plotted.

2.1. The data in A appear different from Figure 2A, although it appears to be the same experiment.

File 1. This figure is difficult to follow.

File 2: It is not clear if the plot in B represents 3 or 7 days after a light pulse.

*Reviewer #2:*

In this manuscript, Henríquez-Urrutia and colleagues establish the existence of a core circadian rhythm in the mycoparasitic fungus Trichoderma atroviride. Using both translational and transcriptional reporters for a core clock component, FRQ/frq, the authors demonstrate that the rhythm meets all the requirements of bona fide circadian oscillator, e.g. entrainment and temperature compensation. Interestingly and importantly, they demonstrate that the robustness of the oscillation is largely dependent on the nutritional environment; specifically, the rhythm was only discernable under growth conditions that likely resemble the natural environment of the fungus. This information will be of keen interest to the chronobiology community, particularly those that attempting to identify a clock in their microbe of interest. Moreover, the authors demonstrate that two processes are under control of the Trichoderma clock, or at least under the regulation of clock components: (1) secondary metabolism, and (2) the myoparasitic interaction with another fungus, Botrytis. Although the connection between those two phenomena in this study is unclear, together they demonstrate the importance of biological clocks in regulating fungal behaviors of interest. Understanding how the time-of-day influences fungal parasitism, for example, could inform the optimization of fungi as biocontrol agents in the agricultural setting. In conclusion, the work provided here is novel, the conclusions are supported by the data, the ideas are well-marshalled. Consequently, I believe this would be would be a good fit for *eLife* and its readership. However, the manuscript may be strengthened by an expanded discussion or additional experiments as outlined below.

1) The authors do an excellent job in defining how environmental parameters (e.g. nutrients and temperature), influence the robustness of Trichoderma core oscillation. It is therefore surprising that the Trichoderma/Botrytis interaction assays were not analyzed across a similar variety of conditions. For example, the authors performed the interaction assays on PDA media at 20C, the data from which support the conclusion the Trichoderma FRQ does not significantly influence its parasitism of Botrytis. However, what would these results look like on media in which the Trichoderma oscillator is more robust, i.e. on GYEC media at 22C? Conversely, how would these changing conditions influence the Botrtyis clock proteins and their role in the interaction? The inclusion of such data may strengthen the overall discussion of the fungal clock operating within stringent environmental conditions.

2) For the interaction assay, it is clear the inoculating the plates at dawn results in greater Trichoderma overgrowth compared to inoculation at dusk. However, the authors state in the discussion (pg 33, line 15), that "overgrowth is increased when the "interactions started at dawn…". Does the inoculation time really represent the time at which the fungal interaction take place? Surely there must be a lag that reflects the growth of the organisms towards each other. Perhaps the interaction begins 12 h after inoculation, i.e. at dusk. Relatedly, how does growth rate of the various mutant strains (of either species) under the various light conditions influence the time in the solar/circadian day in which the initial interaction takes place?

3) Much attention is given to circadian influence on Trichoderma secondary metabolism, which is indeed very interesting. However, the relationship between this and the Botrytis interaction results are fairly weak. Can the authors demonstrate that culture supernatants of Trichoderma taken at different times across the circadian day, or in DD v LL, have an differential impact on Botrytis growth or viability? Maybe this is an oversimplified expectation. In any case , it is noted that Botrytis proteins have the greatest impact on the interaction, which would suggest that the circadian regulation of Trichoderma metabolites may not be particularly important. Do the authors think the relative susceptibility of Botrytis to Trichoderma metabolites, or some other stressor, is regulated by the Bc clock proteins? In other words, some discussion about the fungal-fungal interaction from the Botrytis perspective may be welcomed.

4) Overall, the 'Introduction' and 'Discussion' sections are nicely written and easy to read. By contrast, the 'Results' section contains some phrasing and grammatical issues that should be revised.

This is a very interesting study and my major "concerns" are outlined above. I may have misunderstood key aspects of the experiments, in which case a clarifying statement in the rebuttal may suffice. However, I do think that the authors should clarify some things in the manuscript as outlined here.

1) Please describe the assay conditions more clearly throughout the manuscript. The assay medium for Figure 3, for example, is not mentioned in the results or figure legend. I can't even see a definitive mention of the medium in the methods. In fact, the figure 4 legend is the only place where PDA is explicitly stated. Similarly, it is unclear what media was used for the secondary metabolite profiles. Based on the language in the methods, it seems that the light experiments were performed on PDA and the circadian time course was on GYEC… is that correct? If so, why use different media? This conditions be clearly stated in the figure legends in any case.

2) To what extent does the relative growth rate of the two organisms influence the interaction? For example, if a particular B. cinerea (Bc) mutant were slow growing, would this make the apparent overgrowth of T. atroviride (Ta) seem larger? The methods state that the readout is based on percent overgrowth, but this isn't really obvious in most of the pictures. To me, and perhaps to many readers, the readout (dotted line) seems to reflect the point of contact between the two organisms, sort of like a mating assay, which would presumably be growth rate dependent.

– To perhaps clarify this issue, the authors might consider is showing the each strain (Ta and Bc) by itself on a plate under each of the conditions (e.g. DD, LL, LD, etc). This would allow the reader to better visualize the morphology and growth rate of each organism in isolation, and consequently interpret the interaction experiments better.

3) Pg. 13, line 12: "After 72 h of cultivation in LL, conidia in Δtafrq exhibited a dark-green color compared to the TaWT and the OE::tafrq strains (Supplementary file 2A) which displayed the characteristic T. atroviride green coloration."

– This sentence is a bit confusing. So "green" is normal, and "dark green" is abnormal? It may be helpful for the unfamiliar reader if the authors briefly describe basic Trichoderma colony morphology, i.e. what color are the condia and underlying hyphae.

4) Some specific areas in the 'Results' section that need revision include, but are not limited to, the following:

– Pg 13, line 8: "This distinctive phenotype…"

– Pg. 14, Line 15.

– Page 18, Line 2: "Regarding…"

5) Pg. 7, Line 14: "ostensibly" doesn't seem like the correct word here.

*Reviewer #3:*

The most obvious change during the course of a day is the dark-light cycle. Associated to light changes are changes in harmful UV light, in temperature, in humidity, the possibility for finding food or the orientation in the environment. Circadian clocks help organisms to adapt to these predictable periodic changes before they actually occur. E.g. for humans it was good to wake up during dusk and be alert before their predators are active. For microorganisms, like filamentous fungi, environmental parameters may be important for instance if they produce spores for distribution. It may be advantageous to produce them early in the morning, when the humidity is high, because spores are often produced at the surface of substrates and hence the mycelium is at risk for desiccation. In the case of plant-pathogenic fungi it may be advantageous to spread the spores at times where plants are more susceptible for a fungal attack. This had been shown before by the same research group in the case of Botrytis cinerea. The current paper is a continuation of the work on organismic interactions and highlights the role of a circadian clock in the mycoparasitic relationship between Trichoderma atroviride and the grey mold B. cinerea. They identified a circadian clock in T. atroviride and show that it influences spore and secondary metabolite production. In the mycoparasitic interaction the clock of the prey is important for the success of the intruder rather than the other way around. The downstream components of the circadian clock, which are important for the outcome of the mycoparasitic interaction, remain to be discovered.

The paper has many important implications. The molecular mechanism of circadian clocks has been studied extensively in *Neurospora crassa*, but only in a few other fungi so far. Many fungi seem to lack one of the central components of the *N. crassa* clock, namely the negative element frequency. Therefore, it is important to collect more examples for circadian mechanisms in fungi. In T. atroviride the clock mechanism appears to be conserved in comparison to *N. crassa*. A second important aspect is that T. atroviride is used as a biocontrol agent and hence the work may help to optimize the application of the fungus in the field. Last but not least, the paper shows that secondary metabolism is largely influenced by the clock, an aspect so far largely neglected. Given that filamentous fungi are important producers of secondary metabolites, such as penicillin or lovastatin, circadian clocks should be considered when optimizing production processes.

The first two chapters describe the characterization of the circadian clock of T. atroviride. The authors show that it can be entrained by light-dark cycles, that it can be reset by a light or temperature pulse, and it is temperature compensated in a range between 20 and 26{degree sign}C. The mechanism of clock functioning appears to be similar to the one in *N. crassa* with a conserved cis element in the promoter of the frequency gene. Interestingly, addition of pea powder significantly enhanced the amplitude of the clock-dependent reporter expression. Is this a specific effect of pea or do other plants or fruits serve the same function? This is especially important because this aspect is also highlighted in the Discussion.

I find it very surprising, but interesting at the same time, that several tested and known clock-controlled genes apparently are not clock-controlled in T. atroviride. The authors should look at some regulators of the pathways they identified as being under clock control. The candidates could be further analyzed for the identified cis element in their promoters.

Next, they tested a function of the frq gene outside the clock function, because such an additional function had been described before in B. cinerea. They found that indeed conidiation is delayed in a frq-deletion strain, independent of ligh-dark cycles. To verify these results the deletion strain should be re-complemented with a wild-type copy.

In the following chapters they focus on the mycoparasitic interaction and use frq and wc mutant strains of both fungi in confrontation assays. They first found that light inhibits the parasitic capacity of T. atroviride and that the WC-1 orthologue is required for the aggressive behavior of T. atroviride in the dark. Again in this series of experiments a T. atroviride frq re-complemented strain is missing. In light-dark cycles it turned out that the B. cinerea Frq protein is important for the outcome of the interaction. This shows that the interaction is clock-controlled, but on the prey side.

In the last chapter they analyzed the production of different secondary metabolites and found that the production is under light control. The WC orthologue and Frq are involved in the regulation, independent of their functions in the circadian clock. However, the authors also show that the production of some compounds is clock controlled, and most required Frq. I guess for specialists the source data tables are useful. I cannot evaluate them.

---

## [Author Response]

Essential revisions:The reviewers have made several constructive suggestions that you should carefully consider and address in a point-by-point reply. In particular, please pay particular attention to the following revisions:

We appreciate the editor's comments, summarizing the primary considerations as well as the different comments raised by all reviewers. We provide a point-by-point rebuttal, and hope that now the manuscript meets all criteria for publication.

1) Please perform experiments that demonstrate entrainment (as suggested by Reviewer #1).

While we understand that Reviewer 1 requested this type of experiments to be fully convinced of the presence of a *bona fide* circadian clock, due to technical issues and additional complications associated with the extended pandemic, we could not perform those suggested experiments. Indeed, experimental work was severely hampered at our universities during most of 2021, as access was extremely restricted. Nevertheless, we generated new data that actually go towards that direction, providing additional proof that this is a proper circadian system (see response #1 to Reviewer 1). We truly appreciate the reviewer’s comments as they allowed us to strengthen the dataset regarding this point. We are convinced that the new results not only do that, but also provide new interesting observations included in the manuscript.

2) Please perform the interaction assays across a range of environmental conditions (nutritional, temperature) to determine if the contributions of the clock proteins (in either organism) change accordingly (as suggested by Reviewer #2).

In the new text we now provide an important technical aspect of the confrontation experiments, commenting that only PDA media provides proper conditions to clearly visualize overgrowth or Trichoderma mycelia over Botrytis, as in other media (such as GYEC) Botrytis behaves oddly in terms of mycelia consistence and coloration. Importantly, the main purpose of these experiments was not to systematically test the effect of different conditions on the interaction but instead, to provide a proof of concept that two variables – light (Figure 3), and time (Figure 4) – have a relevant effect modulating the outcome of the interaction. We anticipate that future work in our lab (and the labs of others) will examine the additional effects of other variables in combination with the abovementioned ones.

3) Please re-complement your mutants (as suggested by Reviewer #3).

We thank the reviewer for raising this observation. Accordingly, we have now included complementation experiments as depicted in (new figures: Figure 3—figure supplement 3 and 4) confirming that, as expected, one of the hallmark phenotypes of the mutant is reversed in the complemented strain. We have included information of the new experiments in the Materials and methods section.

Reviewer #1:The goal of this study was to establish that the fungus Trichoderma atroviride, and important biocontrol agent, has a circadian clock, and to test if this clock plays a role in development, secondary metabolite production, and mycoparasitism. The authors clearly establish rhythms in the organism that are dependent on TcFRQ, a homolog of the core clock component FRQ in *N. crassa*, and show that TcFRQ and BRL1, the homolog of the blue light photoreceptor and core clock component WC-1 in *N. crassa*, play differing roles in development, metabolite levels, and mycoparasitism in laboratory conditions. However, the authors do not fully establish that the rhythms are controlled by a circadian clock , nor that TcFRQ is a core clock component. While aspects of this work will be of interest to circadian and fungal biologists, the not all of the aims were met, and the data do not fully support the conclusions.

We sincerely appreciate the positive reception of our work. The reviewer's primary concern is related to the absence of data showing that TaFRQ is indeed a clockcore component. After careful examination, we have pondered our results and performed additional experiments that should help dissipate any potential doubts about that main conclusion. Thus, to fully support the latter, and to clearly establish that a circadian clock controls the observed rhythms, we now have included relevant additional information describing that TaFRQ is indeed a core clock component based on functional complementation experiments (*aka* classic old school approach) in *Neurospora crassa*. Accordingly, as highlighted in the new version of the MS, we do not only demonstrate that *T. atroviride's* clock free runs and compensates for temperature, but also that the *tafrq* gene is capable of functionally complementing a *frq* null strain of *Neurospora* (new figures: Figure 2 and figure supplement 2, and pages 12-13). Most notably, the latter experiment also shows that the “long” free running rhythms characterized in *T. atroviride* can be "transplanted” along with the Tafrq sequence to Neurospora. We think that these data fully support our conclusions, including extra-circadian functions of TaFRQ (see below).

Strengths: The circadian clock in the fungus *N. crassa* serves as an important model for understanding the mechanisms that keep time in eukaryotes, and how this clock controls daily rhythms in gene expression. However, little is known about clocks in other fungi, despite conservation of core clock components found in *N. crassa*, including fungal pathogens and biocontrol agents like Trichoderma. Thus, the strength of this work is in that it goes a long way to establish a clock in T. atroviride that is directly, or indirectly involved in controlling development, metabolism, and mycoparasitism. Importantly, this work opens up the opportunity to determine how the clock impacts fungal-fungal interactions, with the potential of using this to improve biocontrol.

We sincerely appreciate the positive reception of our work.

Weaknesses: There are several weaknesses. The first is that while the authors established conditions to demonstrate two canonical properties needed to establish a circadian clock, a free running rhythm that is close to 24 h and temperature compensation, the authors did not fully test one of the cardinal properties of a circadian clock, entrainment. Second, the authors do not establish that TcFRQ is a core component of the clock oscillator.

We thank the reviewer's thorough review. As mentioned above, although we did not explore entrainment in this work, the new dataset (FRQ complementation) provides new and unequivocal evidence that TaFRQ is a *bona fide* core-clock component, and therefore of the existence of a canonical clock in *Trichoderma*. Additionally, our data also provides clear evidence that the *T. atroviride* system readily responds to temperature and light cues which, albeit it is not entrainment *per se*, is tightly connected with it. Thus, we can now establish that TaFRQ indeed encodes a clock-core component (new figures: Figure 2-figure supplement 2, pages 12 and 13). Having cleared the waters in terms of its clock function, we can also mention about its extra-circadian roles (see following question-answer below). This is important, as we have demonstrated an analogous observation in *Botrytis cinerea*, and we anticipate that more than an exception this extracircadian functions may soon emerge as a transversal and common property of clock components.

Third, the authors made conclusions from a strain that is supposed to overexpress TcFRQ from an actin promoter but, there is no data demonstrating that TcFRQ is actually overexpressed.

Again, we sincerely appreciate the reviewer’s exhaustive review. Based on his/her observations we performed RT-qPCR analysis, which confirmed overexpression, particularly under DD conditions (*circa* 4-fold), info which can now be found in Figure 3figure supplement 2 (new figure) and page 14, Line 21-23. Since we also observed that in LL the degree of overexpression is less marked (albeit still statistically significant), we have made that explicit in the text and in the discussion of the data.

Fourth, the rationale for concluding that TcFRQ has an extra-clock role, as opposed to a clock role, in development and metabolism is not clear.

Indeed, this may have not been clear enough in the text. Now with the new data we have unmistakably stablished that *Tafrq* has a canonical role as a core-clock component. In addition, it has “extra-circadian” roles, as it affects (delays) sporulation and injury response even in LL conditions. As the clock only runs in DD (and not in LL) we refer to these effects as “extra circadian” since they are not based on a running clock.

Finally, a red safety light was used in all circadian experiments in constant dark; however, there is no data to support that T. atroviride is not responsive to red light.

We apologize for the confusion: in all free-running experiments, constant darkness means the total absence of light. Red safety lights are utilized in the dark room, which houses our light-proof incubators (Percival). Thus, experiments (CCD recordings, growth in Petri plates, etc.) were conducted inside the incubators in full darkness or in the presence of white light (when indicated). Red lights were only used to manipule equipment when required, or when opening incubators. Indeed, cultures were seldom exposed to red light in the experiments conducted in this MS. Related to the general question raised by the reviewer, only a few genes (three) have been detected as red-light regulated in *T. atroviride* after a 5 min red light-pulse (PMID: 17074901). In any case, both light intensity and time of exposure significantly differ from occasional expose to redlight during culture manipulation (i.e., harvesting).

Figure 1 While these data support the existence of a circadian clock in this organism based on free running rhythms and temperature compensation, it is important to provide additional data to demonstrate that the rhythm is entrained, one of the 3 criteria established in the field.a. To demonstrate circadian entrainment, as opposed to a direct response to the environmental cues, specifically light, they need to examine different length LD cycles to demonstrate that the clock entrains to the different cycles. In addition, to demonstrate that an acute light or temperature pulse is resetting the clock, as opposed to masking, a a phase response curve is needed to show advances and delays. It is not clear what phase the light pulse was given in Figure 1D.

As mentioned earlier, we were unable to properly perform entrainment experiments. Instead, we adopted a different approach to provide compelling data regarding the characterization of the Trichoderma FRQ-based clock (see complementation experiments). Future work will, undoubtedly, focus on additional circadiana of this system, particularly phase response curves. Nevertheless, we estimate the revised data set provides sufficient information to support the main conclusions of the current manuscript.

b. In the Supplement to Figure 2, several genes are arrhythmic under conditions in which TAFRQLUC is rhythmic. This is an important control to show, for example, that there is not a rhythm in ATP levels that might be driving the rhythm in the TAFRQLUC reporter. I would like to see this mentioned earlier in relation to Figure 1.

We are glad to know that the reviewer also appreciates this data. Indeed, now we mention this is page 9, lines 1-13.

c. It would be helpful to know circadian time in Figure 1, and of interest to know if the peak time is similar to *N. crassa* FRQ.

The data suggests that the phase of the first peak of TaFRQ expression (as FRQ^LUC^) is similar, albeit slightly advanced, compared to what occurs in *N. crassa* (once converted to circadian time), being roughly CT 11 for the latter, and CT 8 for Trichoderma. This is now mentioned in page 7, line 18.

Figure 2 demonstrates that the cis-acting promoter element of *N. crassa* frq that bound by the positive element, the WCC, is necessary for frq mRNA rhythms and functioning of the feedback loop, is sufficient to drive rhythms in luciferase in T. atroviride in DD, and is responsive to light and temperature pulses. This experiment suggests that there is conservation of the c-box, and raises the possibility that TAFRQ functions similarly in the T. atroviride clock as a negative element; however, this experiment raises more questions than it answers, and therefore seems premature. There are several instances in the paper where TaFRQ is called a core clock component, but there are no data to support this claim.

Indeed, this experiment raises a lot of interesting new questions. Nevertheless, in addition to the new complementation data, it provides a general and valuable conclusion: there is conservation of core-clock mechanisms and components across two species belonging to the same class, and there is cross-class recognition of such components. Thus, we believe it nicely contributes to the story.

a. Is there a similar sequence present in the promoter of TAFRQ? If so, what happens if this sequence is mutated?b. What happens to the rhythm, levels, of TAFRQ in a Blr1 deletion or mutated strain?c. Additional experiments are needed to show that Blr1 binds to this sequence and to fully describe TAFRQ role in the clock mechanism.

This reviewer raises interesting observation/questions, that also intrigue us. While these formal questions are highly relevant, we deem that they are not essential in the context of the overall message of this manuscript. Indeed, we have not identified a defined "endogenous" c-box sequence in Trichoderma, nevertheless, as described in the manuscript, “Two putative light response elements in the tafrq’s promoter with a putative GATA box consensus sequence at proximal and distal locations (-355 to -319 and -1180 to -1171 from the TSS, respectively) have been postulated (Cervantes-Badillo et al., 2013), of which one of them could be acting as a putative _Ta_cbox sequence.”.

We did not directly determine TaFRQ levels in a BLR1 mutant strain. Importantly, transcriptomic datasets address this, confirming that, as expected, *tafrq* expression depends on BLR1, and is downregulated in the absence of this TF (PMID: 27020152).

This is now mentioned in Page 4, line 16.

In addition, the collective evidence of the new data showing how TaFRQ complements the Neurospora's *frq* null mutant, the fact that c-box of *Neurospora* works rhythmically in *T. atroviride*, and the absence of rhythms of other promoter-luc constructs in the latter fungus, all clearly reinforces the idea that we are describing a circadian clock and that TaFRQ is indeed a core clock component along with BLR1, aspects that are now commented in the discussion. Thus, while we appreciate and also share some of the questions raised by the reviewer, we believe they are beyond the current scope of the work.

In Supplementary file 1, 2A, B and C, they examine if TaFRQ has a role in development, and this is supported by the data presented by comparing WT versus TaFRQ deletion strains. They show that conidia formation is delayed in the mutant compared to WT, and decreased following light treatment or injury. In addition, they also examined a construct aimed at overexpressing TaFRQ (OE::TaFRQ) using the actin promoter. No differences were observed between WT and OE::TaFRQ; however, it is difficult to make any conclusions here because there were no experiments demonstrating that TaFRQ was actually overexpressed in this strain.

As commented above, we have now measured *tafrq* expression in the overexpressing strain, confirming that levels are statistically higher in the latter compared to WT, which is particularly marked in DD conditions. Indeed, when it comes to the mentioned phenotypes (conidiation, injury response) the OE strain behaves as the WT. Nevertheless, as evidenced in Figure 5, regarding metabolic profiles, the OE strain shows a pattern quite distinct compared to WT, particularly under DD, when expression levels of *tafrq* in OE versus WT are more pronounced.

Figure 3 examines the ability of T. atroviride to overgrow B. cinerea under different lighting conditions and in mutants as an indicator of mycoparasitic behavior. The key findings were that constant light inhibits overgrowth, and that mutations in blr1 in T. atroviride enhances overgrowth in constant darkness. However, it is not clear how these data and those that follow establish and extra-clock role for TaFRQ in these processes.

We apologize for the confusion, as we did not intend to imply that the data in Figure 3 relates to the extra circadian role of TaFRQ. To provide more context, we were also interested to analyze whether TaFRQ had additional roles, besides its clock function, a thought inspired from our findings for Botrytis FRQ. This is a relevant observation since, in the case of Neurospora, the FRQ protein seems to have only clock-related functions, while in Botrytis, in an environmental condition where the clock is not functional (in constant light), the *frq* mutant shows additional extra-circadian functions related to altered development (conidiation). Now, this is better explained in the text in page 15, between lines 09-16.

Figure 4 looks at dawn and dusk confrontations between T. atroviride and B. cinereai in LD cycles. They found that WT T. atroviride and TaFRQ deletion strains had overgrowth that was enhanced at dawn, whereas deletion of TaBLR1 did not show a time of day difference. However, when frq was deleted in B. cinerea, all T. atroviride strains overgrew. These data supported that homologs of *N. crassa* clock components in T. atroviride and B. cinereai play differing roles in time of day specific interactions, and this information could prove useful in biocontrol measures in the future.

We appreciate the reviewer's comments as they highlight our results and potential future applications. Indeed, we are looking forward to see how our observation can be translated to applied studies.

Figure 5 examines if T. atroviride FRQ and BLR1 mutants alter secondary metabolism by examining diffusible and volatile compounds in DD and LL, and following a light pulse. They found several differences, but were unable to correlate these with the overgrowth experiments in Figures 3 and 4. A circadian metabolic profile demonstrated that some compounds were rhythmic in WT and arrhythmic in taFRQ deletion cells. These data provide evidence to support a role for a clock in controlling metabolism in T. atroviride.

Again, we appreciate the reviewer's comments as they highlight the relevance of our results.

Discussion: The paper lacks definitive experiments to show that taFRQ is a core clock component, and therefore, it is not clear how these data support a extra-circadian role for taFRQ. In addition, aspects of the discussion are very lengthy, especially the section describing how this work could foster looking for clocks in other fungi.

We acknowledge, as the reviewer pointed out, the limitation that our study had. Therefore, we took the approach of complementing an arrhythmic Neurospora strain with tafrq (see above). Such results not only provide compelling evidence of TaFRQ role in the clockworks, but also opens the door to interesting future comparative studies. Having dissipated such doubts, it may now be easier to visualize the concept of “extracircadian” roles of FRQ, as they lead to phenotypes even in a condition where clock function is abrogated (constant light). We have also modified parts of the discussion, to make it more fluid.

Methods:a. It is critical to show that T. atroviride is not sensitive to red light given red safety lights were used in circadian experiments.

We have now clarified that low level red-lights are used only to manipulate cultures, equipment, although most of the experiments described herein were conducted in full DD (inside incubators), which were actually housed inside the dark room equipped with the safety lights. Please, refer to (above).

b. The constructs used to examine potential rhythms in known fungal ccgs used the promoter, but not the 3' ends of the genes. The authors may want to consider adding this in the future.

We thank the reviewer's suggestions. Indeed, we are considering this in other studies conducted with collaborators.

Supplemental Figures1.4 In part A, the light pulse did not seem to improve the rhythm similar to what was shown in Figure 1. What the light pulse given at the same time of day?

We have modified the legend of figure 1.4, to better reflect that the conditions of the light pulse were not equivalent to the one in Figure 1.

1.5. When was the light pulse given? These data should be plotted.

We have also modified the figure legend to better explain when the LP was given.

2.1. The data in A appear different from Figure 2A, although it appears to be the same experiment.

We have simplified the legend of figure 2.1 (Figure 2—figure supplement 1) so it is now clear that the experiment was conducted in GYEC, whereas the one in 2A was conducted in GYEC + peas, which (as seen in previous figures) dramatically improves rhythms.

File 1. This figure is difficult to follow.

A series of modifications were made to the figure and its legend for simplification.

File 2: It is not clear if the plot in B represents 3 or 7 days after a light pulse.

This file (now called Supplementary file 3) has been improved as requested.

Reviewer #2:In this manuscript, Henríquez-Urrutia and colleagues establish the existence of a core circadian rhythm in the mycoparasitic fungus Trichoderma atroviride. Using both translational and transcriptional reporters for a core clock component, FRQ/frq, the authors demonstrate that the rhythm meets all the requirements of bona fide circadian oscillator, e.g. entrainment and temperature compensation. Interestingly and importantly, they demonstrate that the robustness of the oscillation is largely dependent on the nutritional environment; specifically, the rhythm was only discernable under growth conditions that likely resemble the natural environment of the fungus. This information will be of keen interest to the chronobiology community, particularly those that attempting to identify a clock in their microbe of interest. Moreover, the authors demonstrate that two processes are under control of the Trichoderma clock, or at least under the regulation of clock components: (1) secondary metabolism, and (2) the myoparasitic interaction with another fungus, Botrytis. Although the connection between those two phenomena in this study is unclear, together they demonstrate the importance of biological clocks in regulating fungal behaviors of interest. Understanding how the time-of-day influences fungal parasitism, for example, could inform the optimization of fungi as biocontrol agents in the agricultural setting. In conclusion, the work provided here is novel, the conclusions are supported by the data, the ideas are well-marshalled. Consequently, I believe this would be would be a good fit for eLife and its readership. However, the manuscript may be strengthened by an expanded discussion or additional experiments as outlined below.1) The authors do an excellent job in defining how environmental parameters (e.g. nutrients and temperature), influence the robustness of Trichoderma core oscillation. It is therefore surprising that the Trichoderma/Botrytis interaction assays were not analyzed across a similar variety of conditions. For example, the authors performed the interaction assays on PDA media at 20C, the data from which support the conclusion the the Trichoderma FRQ does not significantly influence its parasitism of Botrytis. However, what would these results look like on media in which the Trichoderma oscillator is more robust, i.e. on GYEC media at 22C? Conversely, how would these changing conditions influence the Botrtyis clock proteins and their role in the interaction? The inclusion of such data may strengthen the overall discussion of the fungal clock operating within stringent environmental conditions.

We share the curiosity expressed by the reviewer. Examination of different media modulating the Botrytis-Trichoderma interaction will be a future area of study (particularly involving conditions mimicking soil). Nevertheless, they were beyond the scope of this first report, and due to all the complications of a closed lab during almost 16 months, and people moving on to new jobs, this was further delayed. Based on the reactions of colleagues, as we first presented the data in public in 2022 (Fungal Genetics, and Botrytis meetings), we are convinced that the story, as it is, will be extremely well received and, most importantly, will inspire new angles of studies in different labs.

2) For the interaction assay, it is clear the inoculating the plates at dawn results in greater Trichoderma overgrowth compared to inoculation at dusk. However, the authors state in the discussion (pg 33, line 15), that "overgrowth is increased when the "interactions started at dawn...". Does the inoculation time really represent the time at which the fungal interaction take place? Surely there must be a lag that reflects the growth of the organisms towards each other. Perhaps the interaction begins 12 h after inoculation, i.e. at dusk. Relatedly, how does growth rate of the various mutant strains (of either species) under the various light conditions influence the time in the solar/circadian day in which the initial interaction takes place?

As the reviewer noticed, there must be a lag reflecting the growth of both fungi towards each other. Interestingly, in the case of the Botrytis-plant interaction, which also is influenced by the time of the day as reported by our group years ago, there is also a lag: the infection assays are regularly conducted employing a droplet containing conidia that first needs to develop the infecting hyphae, and later, the appressorium to penetrate the plant tissue. However, differences are observed between dawn and dusk inoculation times irrespective of this delay.

The reviewer, for sure, is concerned that this delay might be too extended in time, explained, for example, by a significant difference in growth rate between the two fungi. Therefore, we controlled for it: after two days of growth, the colony sizes of Trichoderma and Botrytis was measured, and no statistical differences were observed. Please, refer to new figure (Figure 4-figure supplement 1) and page 17, Line 9-17.

3) Much attention is given to circadian influence on Trichoderma secondary metabolism, which is indeed very interesting. However, the relationship between this and the Botrytis interaction results are fairly weak. Can the authors demonstrate that culture supernatants of Trichoderma taken at different times across the circadian day, or in DD v LL, have an differential impact on Botrytis growth or viability? Maybe this is an oversimplified expectation. In any case , it is noted that Botrytis proteins have the greatest impact on the interaction, which would suggest that the circadian regulation of Trichoderma metabolites may not be particularly important. Do the authors think the relative susceptibility of Botrytis to Trichoderma metabolites, or some other stressor, is regulated by the Bc clock proteins? In other words, some discussion about the fungal-fungal interaction from the Botrytis perspective may be welcomed.

The question raised by the reviewer is fascinating and could even have biotechnological implications in using Trichoderma for biocontrol. And although the products secreted by Trichoderma when growing alone in a liquid medium differ from those produced when growing with Botrytis in the same Petri dish, the interpretation of the data would be very complex since the secreted products (proteins, metabolites, etc.) are very stable over time. We have briefly included ideas as the ones mentioned by the reviewer, in the discussion session.

4) Overall, the 'Introduction' and 'Discussion' sections are nicely written and easy to read. By contrast, the 'Results' section contains some phrasing and grammatical issues that should be revised.

Thanks. We have revised the Results section.

This is a very interesting study and my major "concerns" are outlined above. I may have misunderstood key aspects of the experiments, in which case a clarifying statement in the rebuttal may suffice. However, I do think that the authors should clarify some things in the manuscript as outlined here.1) Please describe the assay conditions more clearly throughout the manuscript. The assay medium for Figure 3, for example, is not mentioned in the results or figure legend. I can't even see a definitive mention of the medium in the methods. In fact, the figure 4 legend is the only place where PDA is explicitly stated. Similarly, it is unclear what media was used for the secondary metabolite profiles. Based on the language in the methods, it seems that the light experiments were performed on PDA and the circadian time course was on GYEC… is that correct? If so, why use different media? This conditions be clearly stated in the figure legends in any case.

Thanks for bringing this up. We have now made sure that each key figure is explicit regarding the utilized media. The need to use PDA in some experiment’s and GYEC + peas media in others is discussed within the manuscript.

2) To what extent does the relative growth rate of the two organisms influence the interaction? For example, if a particular B. cinerea (Bc) mutant were slow growing, would this make the apparent overgrowth of T. atroviride (Ta) seem larger? The methods state that the readout is based on percent overgrowth, but this isn't really obvious in most of the pictures. To me, and perhaps to many readers, the readout (dotted line) seems to reflect the point of contact between the two organisms, sort of like a mating assay, which would presumably be growth rate dependent.– To perhaps clarify this issue, the authors might consider is showing the each strain (Ta and Bc) by itself on a plate under each of the conditions (e.g. DD, LL, LD, etc). This would allow the reader to better visualize the morphology and growth rate of each organism in isolation, and consequently interpret the interaction experiments better.

Originally, we asked ourselves questions similar to the ones raised by the reviewer. We explored them by changing the relative position of the inoculation point of one of the confronting fungi yet, we saw results similar to the ones presented in the manuscript. Please, refer to response #2 (above).

3) Pg. 13, line 12: "After 72 h of cultivation in LL, conidia in Δtafrq exhibited a dark-green color compared to the TaWT and the OE::tafrq strains (Supplementary file 2A) which displayed the characteristic T. atroviride green coloration."– This sentence is a bit confusing. So "green" is normal, and "dark green" is abnormal? It may be helpful for the unfamiliar reader if the authors briefly describe basic Trichoderma colony morphology, i.e. what color are the condia and underlying hyphae.

Please, see modification in page 14, lines 12-15.

4) Some specific areas in the 'Results' section that need revision include, but are not limited to, the following:– Pg 13, line 8: "This distinctive phenotype…"– Pg. 14, Line 15.– Page 18, Line 2: "Regarding…"

Reviews are appreciated. All these comments were resolved in the corresponding sections.

5) Pg. 7, Line 14: "ostensibly" doesn't seem like the correct word here.

Modified. Now you can read “quality of the oscillations greatly improved”.

Reviewer #3:The first two chapters describe the characterization of the circadian clock of T. atroviride. The authors show that it can be entrained by light-dark cycles, that it can be reset by a light or temperature pulse, and it is temperature compensated in a range between 20 and 26{degree sign}C. The mechanism of clock functioning appears to be similar to the one in *N. crassa* with a conserved cis element in the promoter of the frequency gene. Interestingly, addition of pea powder significantly enhanced the amplitude of the clock-dependent reporter expression. Is this a specific effect of pea or do other plants or fruits serve the same function? This is especially important because this aspect is also highlighted in the Discussion.

At present, we have not done systematic experiments to evaluate other plant species because this would deviate our efforts in the lab. But indeed, this is a very exciting area to explore, and particularly towards identifying the molecular nature of the plant element(s) that seem(s) to enhance rhythms. Future plans entertain the idea of adding different extracts from plant roots, which is a more "natural" environment for *T. atroviride.* Since, as a group, we have been working longer with *B. cinerea*, we routinely use plant material to stimulate conidiation in the latter fungus, but we also do not know the component(s) that promote this enhanced conidiation in *B. cinerea.*

I find it very surprising, but interesting at the same time, that several tested and known clock-controlled genes apparently are not clock-controlled in T. atroviride. The authors should look at some regulators of the pathways they identified as being under clock control. The candidates could be further analyzed for the identified cis element in their promoters.

Indeed, we were surprised too by their lack of rhythmicity. It is possible that the promoter region used does not contain all the information which is discussed in the manuscript. In any case, a systematic and unbiased strategy (e.g., RNA-Seq) could help indicate, in the future, whether orthologs of known *ccgs* (as well as unexpected ones) are rhythmic in Trichoderma.

Next, they tested a function of the frq gene outside the clock function, because such an additional function had been described before in B. cinerea. They found that indeed conidiation is delayed in a frq-deletion strain, independent of ligh-dark cycles. To verify these results the deletion strain should be re-complemented with a wild-type copy.

We acknowledge the reviewer's suggestions. We have verified our results by performing complementation experiments: please, see new Figure 3—figure supplement 3 and 4, (*Δtafrq* complementation), pages 15, lines 9-15.

In the following chapters they focus on the mycoparasitic interaction and use frq and wc mutant strains of both fungi in confrontation assays. They first found that light inhibits the parasitic capacity of T. atroviride and that the WC-1 orthologue is required for the aggressive behavior of T. atroviride in the dark. Again in this series of experiments a T. atroviride frq re-complemented strain is missing. In light-dark cycles it turned out that the B. cinerea Frq protein is important for the outcome of the interaction. This shows that the interaction is clock-controlled, but on the prey side.

We appreciate the reviewer's suggestions. As the lack of *frq* in Trichoderma does not have a major impact in these assays, we mainly looked at the complemented strain in the context of the previous question (see above).

In the last chapter they analyzed the production of different secondary metabolites and found that the production is under light control. The WC orthologue and Frq are involved in the regulation, independent of their functions in the circadian clock. However, the authors also show that the production of some compounds is clock controlled, and most required Frq. I guess for specialists the source data tables are useful. I cannot evaluate them.

Indeed, the data provides nice evidence of rhythmic metabolites (which depend on TaFRQ). We look forward to unveiling the molecular events controlling their production as well as their identity and physiological role.